# Molecular mechanism of hyperactivation conferred by a truncation of TRPA1

Avnika Bali [1], Samantha P. Schaefer[1], Isabelle Trier [1,2], Alice L. Zhang [1], Lilian Kabeche[1,2] & Candice E. Paulsen [1] ✉

A drastic TRPA1 mutant (R919*) identified in CRAMPT syndrome patients has not been mechanistically characterized. Here, we show that the R919* mutant confers hyperactivity when co-expressed with wild type (WT) TRPA1. Using functional and biochemical assays, we reveal that the R919* mutant co-assembles with WT TRPA1 subunits into heteromeric channels in heterologous cells that are functional at the plasma membrane. The R919* mutant hyperactivates channels by enhancing agonist sensitivity and calcium permeability, which could account for the observed neuronal hypersensitivity-hyperexcitability symptoms. We postulate that R919* TRPA1 subunits contribute to heteromeric channel sensitization by altering pore architecture and lowering energetic barriers to channel activation contributed by the missing regions. Our results expand the physiological impact of nonsense mutations, reveal a genetically tractable mechanism for selective channel sensitization, uncover insights into the process of TRPA1 gating, and provide an impetus for genetic analysis of patients with CRAMPT or other stochastic pain syndromes.

The wasabi receptor, TRPA1 (Transient Receptor Potential Ankyrin subtype 1), is a calcium-permeable non-selective homotetrameric cation channel expressed in a subset of primary sensory neurons of the dorsal root, trigeminal, and nodose ganglia where it plays a role in many pain-associated sensory disorders including itch, airway diseases, and inflammation[1–6]. Outside the sensory system, TRPA1 has also been implicated in respiratory, cardiovascular, gastrointestinal, and muscular physiology[3,7–9]. The TRPA1 chemosensor is activated by a chemically diverse panel of environmental and endogenous toxins that fall into two broad categories: electrophiles and non-electrophiles. Electrophile agonists such as Allyl Isothiocyanate (AITC) and Cinnamaldehyde activate the channel by covalently modifying key conserved cysteine residues in the membrane proximal cytoplasmic N-terminus (Fig. 1a, yellow circles)[10–13]. Non-electrophile agonists such as Carvacrol and 2-Aminoethoxydiphenyl Borate (2-APB) bind at distinct sites in the transmembrane domain to promote channel activation[14–17]. There is also evidence that mammalian TRPA1 channels serve as noxious cold receptors[10,18]. Recent cryo-electron microscopy (cryo-EM) TRPA1 structures captured in the closed and electrophile agonist

activated states reveal large, allosterically coupled conformational changes in the transmembrane domain and membrane-proximal cytoplasmic N- and C-termini, referred to as the allosteric nexus, highlighting key roles for these regions in channel gating (Fig. 1a–c)[19–21].

The TRPA1 transmembrane domain and allosteric nexus account for only 35% of the total protein sequence (390 of 1119 amino acids) and little is known about the functional impact of the remaining cytoplasmic domains. Situated below the allosteric nexus, TRPA1 houses a large N-terminal ankyrin repeat domain (ARD), in which five membrane proximal ARs form a cage around a C-terminal coiled coil that coordinates a requisite polyanion cofactor (Fig. 1a–c)[19,22]. Interestingly, certain TRPA1 species orthologues can be activated by temperature, though the molecular mechanism of temperature sensing is unclear[23]. Studies with TRPA1 species chimeras and point mutations have shown thermosensation can be conferred or tuned through the ARD[24,25]. Whether or how the ARD can communicate to the channel pore remains unknown.

Ion channel mutations with associated pathological conditions, known as channelopathies, provide direct evidence a channel is

[1]Department of Molecular Biophysics and Biochemistry, Yale University, New Haven, CT, USA. [2]Cancer Biology Institute, Yale University, West Haven, CT, USA. ✉e-mail: candice.paulsen@yale.edu

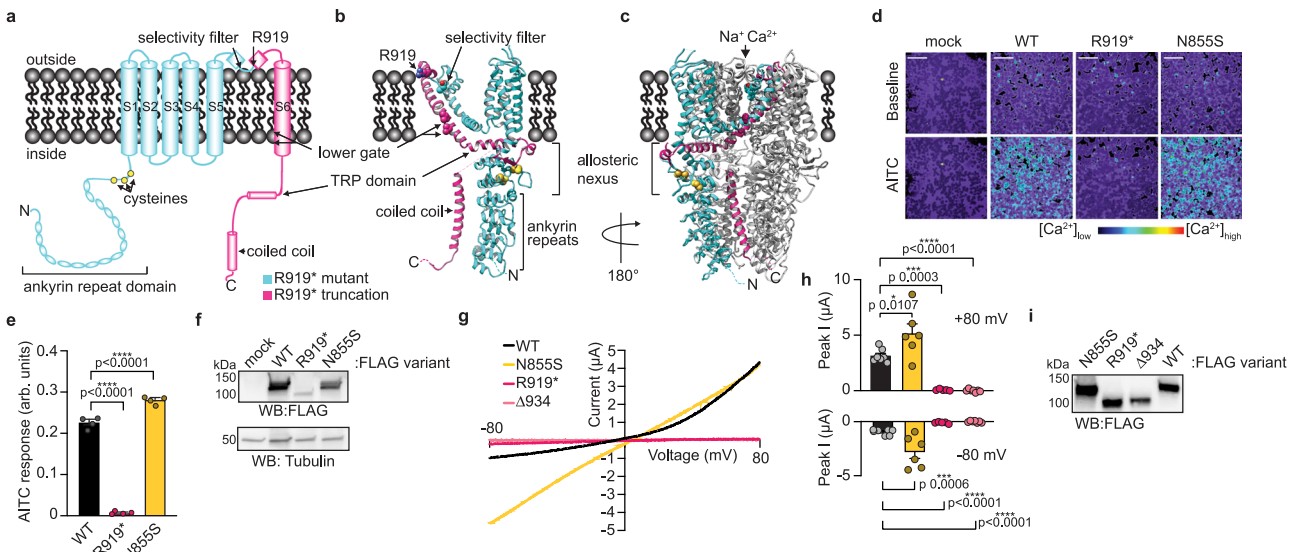

**Fig. 1 | The R919* mutant is a nonfunctional *TRPA1* natural variant. a** Cartoon schematic of a full-length hTRPA1 monomeric subunit, with relevant structural features denoted. Regions retained in R919* hTRPA1 are indicated in teal and regions truncated are indicated in pink. Ribbon diagrams of WT hTRPA1 atomic model for residues K446-T1078 from a single subunit (**b**) and the homotetrameric channel (**c**). Color scheme and relevant structural features denoted as in **a**. In **c**, only one subunit is colored for clarity. Allosteric nexus indicated with brackets. Models built with the human TRPA1 Cryo-EM structure (PDB: 6V9W) in UCSF Chimera. **d** Ratiometric calcium imaging of HEK293T cells transfected with empty vector (mock), WT, R919*, or N855S hTRPA1. Cells were stimulated with 100 μM AITC. Images are representative of three independent experiments. Scale bars indicate 100 μm. **e** Quantification of AITC-evoked change in Fura-2 ratio (arbitrary units, arb. units) for WT (black), R919* (deep pink), or N855S (yellow) hTRPA1 variants. Data represent mean ± SEM. $n = 4$ independent experiments, $n \geq 90$ cells per transfection condition per experiment. ****$p < 0.0001$, one-way ANOVA with Bonferroni's

*post hoc* analysis. **f** Western blot of lysates from cells expressing 3×FLAG-tagged hTRPA1 variants, probed using HRP-conjugated anti-FLAG antibody. Tubulin was the loading control. Results were verified in four independent experiments. **g** Representative voltage ramp (−80 mV to +80 mV) current-voltage (I-V) relationships from *Xenopus laevis* oocytes expressing WT (black), N855S (yellow), R919* (deep pink), or Δ934−1119 (pink) hTRPA1 variants. Currents evoked by 250 μM AITC. Extracellular solution contained no calcium. **h** Quantification of AITC-evoked peak current amplitudes at −80 mV and +80 mV. Colors as indicated in **g**. Data represent mean ± SEM. $n = 5$ (R919* and Δ934−1119 hTRPA1), 6 (N855S hTRPA1), or 7 (WT hTRPA1) independent oocytes. ****$p < 0.0001$, ***$p < 0.001$, *$p < 0.05$, one-way ANOVA with Bonferroni's *post hoc* analysis. **i** Western blot of lysates from representative *Xenopus* oocytes used for recordings in **g** expressing 3×FLAG-tagged hTRPA1 variants, probed using HRP-conjugated anti-FLAG antibody. **f**, **i** Full blots are included in Supplementary Fig. 14. **e**, **g**, **h** Source data are provided as a Source Data file.

relevant to human health, and can illuminate molecular mechanisms and protein domains that govern normal and aberrant channel function[26]. To date, two human *TRPA1* disease mutations associated with pain disorders have been reported, illustrating a direct role for TRPA1 in pain signaling and marking it as a promising therapeutic target[18,27]. The first characterized TRPA1 channelopathy, which was discovered in patients with the rare, autosomal dominant Familial Episodic Pain Syndrome (FEPS), introduced a missense mutation at N855[18]. The N855S TRPA1 mutant forms functional channels when expressed alone in heterologous cells characterized by increased inward current at resting membrane potentials with no change to agonist sensitivity. The location of the N855S mutation in a short linker that physically couples two transmembrane helical bundles highlighted a key role for this region in channel gating years before high-resolution TRPA1 structures were available.

The second reported TRPA1 channelopathy has not yet been characterized and a mechanism for how it may alter channel function is unknown. This mutant was discovered in a father-son pair with cramp-fasciculation and other neuronal hyperexcitability-hypersensitivity symptoms including cold hyperalgesia, chronic itch, and asthma that are consistent with aberrant TRPA1 function[27]. This collection of symptoms was referred to as CRAMPT syndrome[27]. The CRAMPT *TRPA1* variant introduces a nonsense mutation at R919 located at the start of the second pore helix, and the resulting R919* TRPA1 protein lacks the final 201 amino acids, including critical elements involved in gating such as the pore-lining transmembrane helix (S6) and components of the allosteric nexus (Fig. 1a–c). In clinical settings, it is frequently assumed that pathologies linked to truncations of channel proteins originate from haploinsufficiency or dominant negative

effects. However, such broad assumptions have been challenged as more rigorous studies of truncation proteins reveal numerous mechanisms through which they may underlie hyperexcitability-hypersensitivity phenotypes[28–32]. In this study, we seek to elucidate whether and how the R919* TRPA1 mutant influences channel activity that could explain the observed pain phenotype in CRAMPT patients. Our results expand the functional impact of drastic truncation mutants and provide an impetus for whole exome sequencing of patients with seemingly stochastic pain phenotypes.

## Results

### Functional characterization of the R919* TRPA1 mutant

Whole-exome sequencing and co-segregation analysis of a father-son pair with CRAMPT syndrome identified a novel *TRPA1* gene variant consisting of a C to T transition (*c.2755C>T*)[27]. When mapped onto the wild type (WT) human TRPA1 monomer structure, the R919* mutant truncates the last 201 amino acids, or 18% of the protein, removing many important structural and regulatory features including the second pore helix, the ion conduction pathway-lining S6 transmembrane helix and the entire cytoplasmic C-terminus (Fig. 1a–c).

Channelopathies are generally first studied in isolation in heterologous systems to reveal alterations to channel biophysical properties[18,33–36]. For example, the previously reported FEPS-associated TRPA1 mutant (N855S) exhibits enhanced channel activity when expressed in isolation[18]. Since patients carrying the R919* TRPA1 mutant exhibit hypersensitivity-hyperexcitability symptoms, we tested whether R919* TRPA1 inherently possesses any enhanced functional properties. Human R919* TRPA1 was expressed in HEK293T cells or *Xenopus laevis* oocytes, which do not natively express TRPA1, and

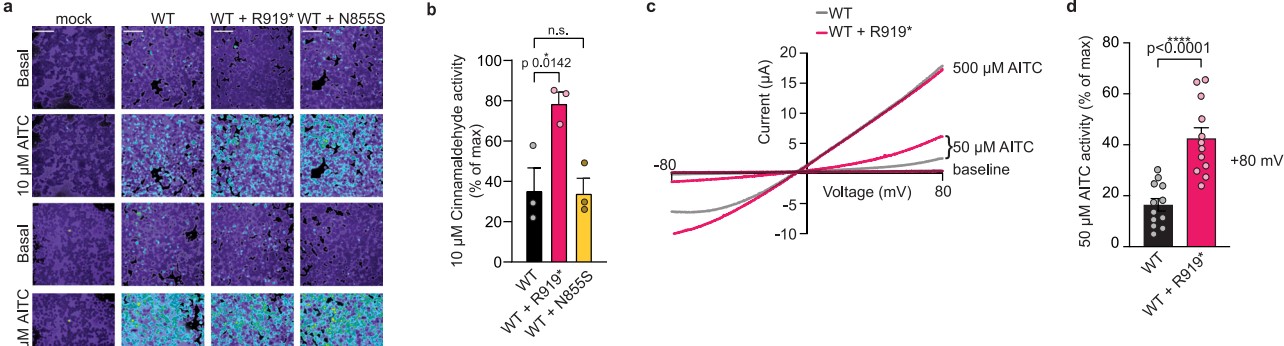

**Fig. 2 | The R919\* mutant confers hyperactivity when co-expressed with WT TRPA1. a** Ratiometric calcium imaging of cells transfected with empty vector (mock), WT, WT and R919\*, or WT and N855S hTRPA1. Cells were stimulated with 10 μM (top) or 100 μM (bottom) AITC. Images are representatives from three independent experiments. Scale bars indicate 100 μm. **b** Quantification of 10 μM Cinnamaldehyde-evoked change in Fura-2 ratio for WT (black), WT and R919\* (pink), or WT and N885S (yellow) hTRPA1 relative to maximum response of each expression condition at 100 μM Cinnamaldehyde. Data represent mean ± SEM. \**p* < 0.05, n.s. not significant. *n* = 3 independent experiments, *n* ≥ 90 cells per

condition per experiment, one-way ANOVA with Bonferroni's *post hoc* analysis. **c** Representative raw I-V relationships from *Xenopus* oocytes expressing WT (black) or WT and R919\* (pink) hTRPA1. Baseline currents and currents evoked by sub-saturating (50 μM) or saturating (500 μM) AITC concentrations are shown. Extracellular solution contained 1.8 mM calcium. **d** Quantification of 50 μM AITC-evoked peak current amplitudes normalized to 500 μM AITC-evoked currents (max) at +80 mV holding potential. Colors as indicated in **c**. Data represent mean ± SEM. \*\*\*\**p* < 0.0001. *n* = 12 oocytes per condition, two-tailed Student's *t*-test. **b**, **d** Source data are provided as a Source Data file.

channel activity was assayed by Fura-2 ratiometric calcium imaging or whole-cell voltage clamp recordings, respectively. HEK293T cells and *Xenopus* oocytes expressing the R919\* TRPA1 mutant revealed no activity in the presence of either the electrophile agonist AITC (Fig. 1d, e, g, h) and the non-electrophile agonist Carvacrol (Supplementary Fig. 1a), compared to the robust activation observed for WT TRPA1. Additionally, the N855S TRPA1 mutant exhibited enhanced channel activity with a loss of channel rectification and a 3.8-fold increase in inward currents in *Xenopus* whole-cell recordings consistent with its reported gain-of-function properties (Fig. 1d, e, g, h, and Supplementary Fig. 1a)[18]. Immunoblot analysis confirmed R919\* mutant protein was produced in HEK293T cells and *Xenopus* oocytes, albeit at significantly lower levels than WT or N855S TRPA1 in HEK293T cells (Fig. 1f, i). Nonetheless, WT TRPA1 still formed functional channels when expressed at a comparable amount to R919\* mutant (Supplementary Fig. 1b, c). These results suggest that the lack of inherent activity for R919\* TRPA1 is due to its inability to produce functional calcium permeable ion channels, which is unsurprising given the loss of the many key structural and functional domains from this mutant (Fig. 1a–c).

### Functional characterization of WT and R919\* TRPA1 co-expression

Both R919\* and N855S TRPA1 mutants were discovered in heterozygous individuals, where WT and mutant TRPA1 protein could be co-produced[18,27]. This raised the possibility that the R919\* mutant mediates its effects by altering co-expressed WT TRPA1 properties. To test whether the R919\* mutant affected channel activity in the presence of WT TRPA1, ratiometric calcium imaging was performed on HEK293T cells co-expressing both variants. Cells co-expressing WT and R919\* TRPA1 exhibited a robust increase in calcium influx compared to cells expressing WT TRPA1 alone, especially in response to sub-saturating concentrations of AITC (Fig. 2a and Supplementary Fig. 2a), Cinnamaldehyde (Fig. 2b), and Carvacrol (Supplementary Fig. 2b, c). A similar robust increase in whole-cell currents was observed with a sub-saturating AITC concentration in *Xenopus* oocytes co-expressing WT and R919\* TRPA1 compared to those expressing WT protein alone (Figs. 2c, d and 3a, b). These channels were inhibited by the canonical TRPA1 antagonists A-967079, HC-030031, and ruthenium red whether they were applied before or after channel activation, consistent with TRPA1-specific responses in both functional assays

(Fig. 3a, b and Supplementary Fig. 3a–d). Patients harboring the R919\* *TRPA1* mutation experienced symptom relief with the sodium channel blocker carbamazepine (CBZ) raising the possibility that this molecule directly inhibits TRPA1 channels; however, CBZ failed to affect TRPA1 activity at concentrations below 1 mM (Supplementary Fig. 3e–h). Together, these observations suggest that co-expression of WT TRPA1 with the R919\* mutant yields hyperactive channels that are sensitive to canonical TRPA1 antagonists but not to CBZ, which likely abolishes pain responses downstream of TRPA1.

Enhanced channel activity at sub-saturating AITC concentration hints that co-expression of WT and R919\* TRPA1 affects agonist sensitivity. This is distinct from cells co-expressing WT and N855S TRPA1, which displayed a clear enhancement of calcium influx at both sub-saturating and saturating AITC concentrations (Fig. 2a). While the N855S mutant exhibits increased calcium influx at each agonist concentration, its agonist sensitivity was previously shown to be no different from WT TRPA1 with the electrophile agonist Cinnamaldehyde[18]. Cinnamaldehyde sensitivity was quantified by determining the half maximum effective concentration (EC$_{50}$) from dose-response curves of cells expressing WT TRPA1 alone or with the N855S or R919\* mutants. While no significant change in the Cinnamaldehyde EC$_{50}$ was observed when WT TRPA1 was co-expressed with the N855S mutant (37.7 μM for WT TRPA1 versus 43.1 μM for WT and N855S TRPA1), a 4.8-fold increase in Cinnamaldehyde sensitivity (EC$_{50}$ = 7.8 μM) was observed with R919\* TRPA1 co-expression (Fig. 3c). Significantly enhanced agonist sensitivity was also observed with Carvacrol, indicating the R919\* mutant mediates channel hypersensitivity independent of agonist type (Supplementary Fig. 2d).

Collectively, these results reveal that the R919\* mutant confers enhanced agonist sensitivity when co-expressed with WT TRPA1, consistent with mutant-mediated sensitization and hyperactivation of TRPA1 channels (Figs. 2, 3 and Supplementary Fig. 2). In contrast, the N855S mutant has no effect on agonist sensitivity. Thus, the mechanism of channel hyperactivity conferred by the R919\* mutant is likely distinct from N855S TRPA1.

### Effect of the R919\* mutant on WT TRPA1 expression, localization, and general cell stress

Channels can become hyperactive through altered protein stability, enhanced plasma membrane trafficking or via post-translational channel modifications triggered by general cell stress[37–47].

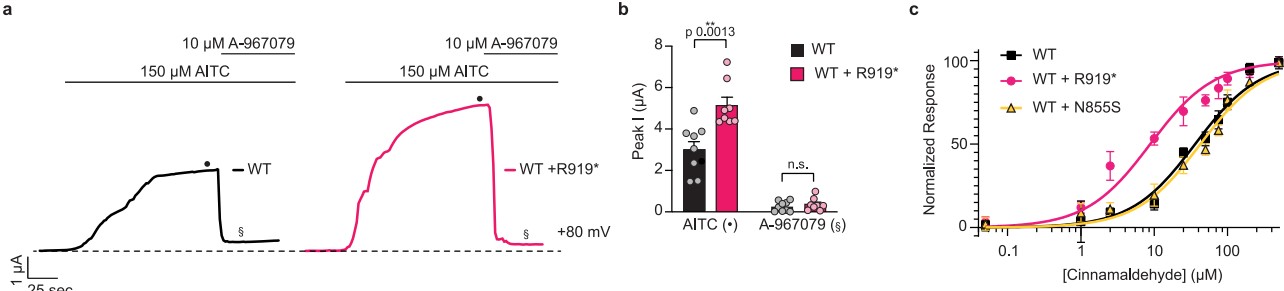

**Fig. 3 | Hyperactive channels retain antagonist sensitivity and gain enhanced agonist sensitivity. a** Representative whole-cell voltage-clamp recordings of *Xenopus* oocytes expressing WT hTRPA1 (black) or WT and R919* TRPA1 (pink) at +80 mV holding potential. Currents evoked by 150 µM AITC (black dot) and inhibited by 10 µM A-967079 (double s). Dashed line denotes 0 µA current. Protocol of condition application indicated above. Extracellular solution contained no calcium. **b** Quantification of AITC-evoked and A-967079-inhibited peak current amplitudes at +80 mV. Colors as indicated in **a**. Data represent mean ± SEM. **p < 0.01, n.s. not significant. *n* = 8 oocytes per condition, two-tailed Student's

*t*-test. **c** Dose-response curve of Cinnamaldehyde-evoked calcium responses for cells transfected with WT (black), WT and R919* (pink), or WT and N855S (yellow) hTRPA1. Calcium responses normalized to maximum calcium response to 500 µM Cinnamaldehyde. Traces represent average ± SEM of normalized calcium responses from 3 independent experiments, *n* = 30 cells per agonist concentration per experiment. Data were fit to a non-linear regression. EC$_{50}$ (95% CI) values are 37.7 µM for WT hTRPA1 (95% CI, 30.5–45.8 µM), 7.8 µM for WT and R919* hTRPA1 (95% CI, 6.1–9.9 µM), and 43.1 µM for WT and N855S hTRPA1 (95% CI, 33.1–54.5 µM). **b**, **c** Source data are provided as a Source Data file.

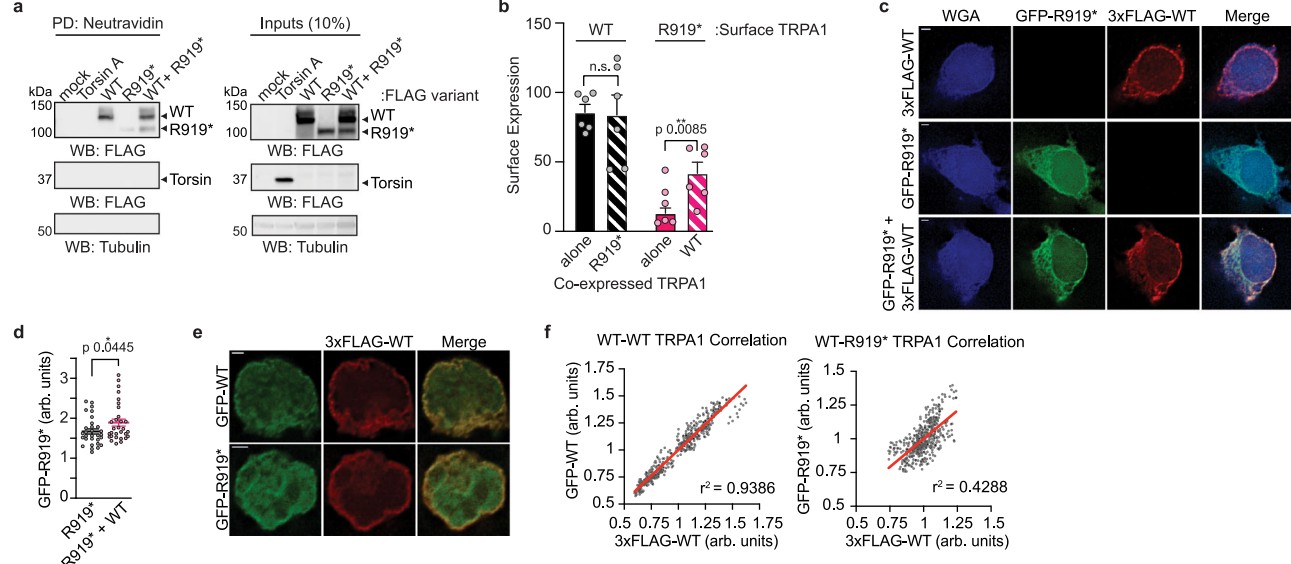

**Fig. 4 | WT and R919* TRPA1 co-localize at the plasma membrane in cells.**
**a, b** Surface biotinylation analysis of 3×FLAG-WT or R919* hTRPA1 or FLAG-Torsin A from transfected HEK293T cells. Biotinylated proteins were precipitated by Neutravidin resin pulldown and probed using HRP-conjugated anti-FLAG antibody (**a**) and quantified (**b**). Tubulin from whole cell lysates (10%, inputs) was the loading control. Torsin A and Tubulin were negative controls for relative plasma membrane localization. Full blots are included in Supplementary Fig. 14. **b** Expression of WT (black) or R919* (pink) hTRPA1 in the plasma membrane is represented by filled bars (corresponding to lanes 2 and 3 in **a**, respectively). Expression of WT or R919* hTRPA1 in the plasma membrane from co-transfected cells is represented by striped bars (corresponding to lane 4 in **a**). Data represent mean ± SEM. **p < 0.01, n.s. not significant. *n* = 6 independent experiments, two-tailed Student's *t*-test. **c** Representative deconvolved immunofluorescence images of HEK293T cells expressing GFP-R919*, 3×FLAG-WT, or GFP-R919* and 3×FLAG-WT hTRPA1. Cells were stained with anti-GFP (green) and anti-FLAG (red) antibodies. Plasma

membrane was labeled with wheat germ agglutinin (blue). Scale bar indicates 2 µm. Images are representative of 3 independent experiments. **d** Quantification of GFP-R919* hTRPA1 fluorescence intensity in the plasma membrane of HEK293T cells relative to the cell interior (arbitrary units, arb. units). Values were obtained using line-scans of raw images. Data represent mean ± SEM. *p < 0.05, *n* = 30 cells per condition, two-tailed Student's *t*-test. **e** Representative deconvolved immunofluorescence images of HEK293T cells co-expressing 3×FLAG-WT hTRPA1 with GFP-WT or GFP-R919* hTRPA1. Cells were stained with anti-GFP (green) and anti-FLAG (red) antibodies. Scale bar indicates 2 µm. Images are representative of 3 independent experiments. **f** Scatter plot of 3×FLAG-WT hTRPA1 and GFP-WT (left) or R919* hTRPA1 (right) fluorescence intensity at the plasma membrane of HEK293T cells. Pearson's correlation coefficients (r) were determined using raw images and the coefficient of determination (r$^2$) is depicted in the lower right corner of each plot. A line of best fit is shown in red. *p < 0.0001, *n* > 450 pixels in 1 cell per condition. **b**, **d**, **f** Source data are provided as a Source Data file.

Immunoblot analysis indicated that expression of R919* had no effect on WT TRPA1 levels, even when performed on cells exhibiting hyperactivity (Supplementary Fig. 4a–c). Similarly, individual or co-expression of WT and R919* TRPA1 had no overt effects on cell viability, reactive oxygen species production, or ER stress induction (Supplementary Fig. 4d–f). It is possible that R919* TRPA1 causes cell stress not detected by these assays that is capable of generally hypersensitizing ion channels. Thus, R919* TRPA1 was co-expressed

with the related capsaicin receptor, TRPV1 and assayed for its effect on human TRPV1 expression, activity, or agonist sensitivity. Akin to WT TRPA1, the R919* mutant had no effect on TRPV1 expression (Supplementary Fig. 4a, b). Moreover, co-expression with R919* TRPA1 conferred no change in capsaicin evoked TRPV1 calcium influx or agonist sensitivity (Supplementary Fig. 5). Finally, surface biotinylation assays revealed that co-expression of R919* had no effect on the amount of WT TRPA1 at the plasma membrane (Fig. 4a, b,

compare solid and striped black bars). Together, these data indicate that the R919* mutant does not sensitize channels via changes in WT TRPA1 expression, altered plasma membrane localization, or general induction of cell stress.

### Effect of WT TRPA1 on R919* mutant expression and localization

Some but not all ion channel truncation mutants result in defects in plasma membrane trafficking[28,42,46,48–52]. Surface biotinylation assays revealed a small plasma membrane population of the R919* mutant, which increased significantly when co-expressed with WT TRPA1 (Fig. 4a, b, compare solid and striped pink bars). Overexpressed Torsin A, an endoplasmic reticulum and perinuclear space-resident AAA+ ATPase, was used as an internal negative control to ensure the R919* mutant labeling was attributed to true plasma membrane localization and not to probe internalization during such assays (Fig. 4a)[53]. We further investigated the R919* mutant plasma membrane localization using immunofluorescence imaging. Deconvolved images of HEK293T cells expressing GFP-tagged R919* TRPA1 or 3×FLAG-tagged WT TRPA1 revealed robust WT TRPA1 localization at the cell surface, while the R919* mutant displayed aberrant localization spreading more diffusely throughout the cytoplasm with minimal plasma membrane localization (Fig. 4c). When co-expressed, 3×FLAG-tagged WT and GFP-tagged R919* TRPA1 both co-localized at the plasma membrane (Fig. 4c). Cross-sectional line scans were performed on raw images of immuno-stained cells, revealing a statistically significant increase in R919* mutant at the cell surface when co-expressed with WT TRPA1 (Fig. 4d and Supplementary Fig. 6a, b). Notably, co-expression with WT TRPA1 did not increase the R919* mutant protein expression, suggesting enhanced plasma membrane localization is not simply due to stabilization of R919* TRPA1 protein (Fig. 4a and Supplementary Fig. 4a, b).

Interestingly, deconvolved immunofluorescence images of HEK293T cells co-expressing GFP-tagged R919* TRPA1 and 3×FLAG-tagged WT TRPA1 exhibit apparent co-localization of WT and R919* TRPA1 at the plasma membrane (Fig. 4c, e). To further assess the degree of co-localization between WT and R919* TRPA1 at the plasma membrane, line scan analysis of plasma membrane segments was performed on raw images of immuno-stained cells[54,55]. Such analysis on HEK293T cells co-expressing GFP-tagged WT TRPA1 and 3×FLAG-tagged WT or R919* TRPA1 revealed a strong positive correlation between WT subunits that is consistent with known channel homotetramerization, and a positive, albeit weaker correlation between WT and R919* TRPA1 signal at the cell surface (Fig. 4f and Supplementary Fig. 6c–e). Together, these data suggest that co-expression of WT and R919* TRPA1 influences the localization of the R919* mutant, and that both co-localize at the plasma membrane.

### Physical interaction of WT and R919* TRPA1 in cells

A direct interaction between WT and R919* TRPA1 would explain their observed co-localization in HEK293T cells (Fig. 4) and could structurally impact WT TRPA1 to confer the observed channel sensitization (Fig. 2). To test whether WT and R919* TRPA1 directly interact, pull-down assays were conducted on lysates from cells co-expressing differentially tagged variants of WT or R919* TRPA1. These pulldown assays revealed a robust interaction between MBP-tagged WT TRPA1 and 3×FLAG-tagged WT and R919* TRPA1, but not with 3×FLAG-tagged Kv1.2/2.1 (Kv), a voltage-gated potassium channel (Fig. 5a). Similarly, MBP-tagged R919* TRPA1 was able to efficiently pulldown 3×FLAG-tagged WT and R919* TRPA1, with no interaction observed with 3×FLAG-tagged Kv (Fig. 5a).

A proximity biotin ligation assay was used to further investigate if WT and R919* TRPA1 directly interact in intact cells[56]. BioID2-fused WT TRPA1 or Kv were co-expressed with 3×FLAG-tagged WT TRPA1, R919* TRPA1, or Kv. Biotinylated 3×FLAG-tagged variants were detected by

immunoblot analysis of Neutravidin eluates. These experiments revealed that 3×FLAG-tagged WT and R919* TRPA1 were biotinylated by BioID2-fused WT TRPA1, but not by BioID2-fused Kv (Fig. 5b). Additionally, 3×FLAG-tagged Kv was biotinylated by BioID2-fused Kv, but not by BioID2-fused WT TRPA1 (Fig. 5b). Collectively, our results indicate that WT and R919* TRPA1 engage in a close-range physical interaction in cells.

### Biochemical characterization of isolated WT and R919* TRPA1 complexes

The proximity biotinylation results reveal that WT and R919* TRPA1 subunits are within a range of ~10 nm, approximately the width of a single TRPA1 channel[19]. This labeling radius is consistent with proximity of adjacent WT and R919* TRPA1 homotetramers or with co-assembly of WT and R919* TRPA1 subunits into heteromeric complexes. While TRPA1 is the only member of the TRPA sub-family of mammalian TRP ion channels, heteromerization within the multi-member TRPV, TRPC, TRPP, and TRPML sub-families is well characterized[57–67], raising the possibility that WT and R919* TRPA1 subunits could heteromerize in a similar manner. Indeed, such heteromerization of WT subunits with alternative splice variants or single nucleotide polymorphisms has been proposed for both TRPV1 and TRPA1[68,69].

Fluorescence size exclusion chromatography (FSEC) was used to differentiate between association of adjacent homotetramers versus assembly of heteromeric complexes as these assemblies are of very different sizes (e.g., ~1000 KD versus ~500 KD, respectively) and would yield distinct elution profiles[70]. GFP- or 3×FLAG-tagged WT or R919* TRPA1 were co-expressed in HEK93T cells in specific combinations, complexes were isolated by anti-FLAG immunoprecipitation, and eluates were analyzed by FSEC to monitor the associated GFP-tagged subunits. The elution profile from GFP WT TRPA1 co-purified with 3×FLAG WT TRPA1 revealed a monodisperse peak consistent with a TRPA1 tetramer, as previously reported (Fig. 5c, black trace)[19]. These complexes were of similar size to homotetrameric GFP-WT TRPA1 channels suggesting the heterogeneity of a large GFP tag and a small 3×FLAG tag had minimal impact on the elution profile (Supplementary Fig. 7a). Complexes purified from cells co-expressing GFP- and 3×FLAG-tagged R919* TRPA1 revealed a monodisperse peak that eluted slightly before WT TRPA1 homotetramers (Fig. 5c, teal trace). Interestingly, complexes isolated from cells co-expressing GFP-tagged WT TRPA1 and 3×FLAG-tagged R919* TRPA1 revealed a monodisperse peak that elutes slightly earlier than WT or R919* homotetramers (Fig. 5c, pink trace). Immunoblot analysis of FSEC fractions from the TRPA1 (indicated with a black dot) and free FLAG peptide (indicated with an infinity symbol) peaks confirmed the presence of 3×FLAG-tagged variants in the TRPA1 peaks (Fig. 5d). While fluorescent signal in these experiments was only attributed to one subunit type, multicolor FSEC experiments can be used to characterize heteromeric membrane protein assemblies[71]. Dual color FSEC analysis of Ni²⁺-FLAG tandem-purified complexes from cells co-expressing 8xHis-mCerulean-tagged WT and FLAG-mVenus-tagged R919* TRPA1 revealed an overlapping peak (indicated with a black dot) in both the mCerulean and mVenus channels consistent with TRPA1 peaks in Fig. 5 (Supplementary Fig. 7b, c). Importantly, mCerulean and mVenus have non-overlapping spectral properties (Supplementary Fig. 7d, e), confirming that the TRPA1 peak detected at both wavelengths from tandem-purified samples must contain both TRPA1 variants.

To further interrogate the oligomer state of WT-R919* complexes, Strep-FLAG tandem affinity purification was performed on lysates from HEK293T cells expressing dual-tagged FLAG-MBP- or Strep-MBP-WT or R919* TRPA1 variants and analyzed by Blue Native PAGE. These results also suggest WT-R919* complexes resemble WT TRPA1 homotetramers in size (Supplementary Fig. 7f). Notably, R919* homomers were not resolved after tandem purification by this method, perhaps suggesting

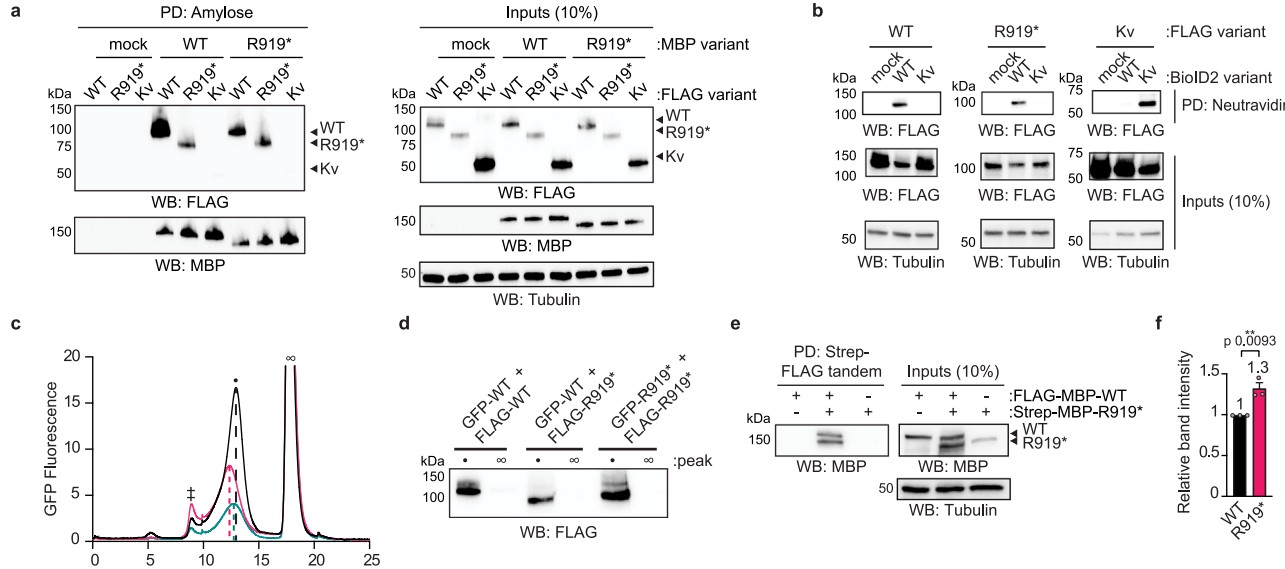

**Fig. 5 | WT and R919\* TRPA1 associate in cells to form complexes of similar size to homotetrameric WT channels. a** Immunoblotting analysis of 3×FLAG-WT hTRPA1, R919\* hTRPA1, or Kv1.2/2.1 protein expression after amylose pulldown from lysates of cells co-transfected with empty vector (mock), MBP-WT hTRPA1, or MBP-R919\* hTRPA1. Samples were probed using HRP-conjugated anti-FLAG antibody. MBP-WT and R919\* hTRPA1 were probed using anti-MBP antibody. Blots representative of three independent experiments. **b** Immunoblotting analysis of biotinylated 3×FLAG-WT hTRPA1, R919\* hTRPA1, or Kv1.2/2.1 co-expressed with empty vector (mock), BioID2-WT hTRPA1, or BioID2-Kv1.2/2.1. Biotinylated proteins were precipitated by Neutravidin resin pulldown and probed using HRP-conjugated anti-FLAG antibody. Blots representative of three independent experiments. **c** FSEC chromatograms from cells co-expressing GFP-WT and 3×FLAG-WT hTRPA1 (black), GFP-WT and 3×FLAG-R919\* hTRPA1 (pink), or GFP-R919\* and 3×FLAG-R919\* hTRPA1 (teal). FLAG immunoprecipitated eluates were analyzed by FSEC. Chromatograms reveal elution profiles of co-purified GFP-WT or R919\* hTRPA1 complexes. Peaks corresponding to void (double cross), tetrameric WT hTRPA1 channels (black dot)

and free FLAG peptide (infinity symbol) are indicated. Dashed lines denote the center elution volume of each co-purified complex. Results were verified in 3 independent trials. **d** Immunoblotting analysis of 3×FLAG-WT or R919\* hTRPA1 from indicated peak fractions from (**c**) probed using HRP-conjugated anti-FLAG antibody. **e** Immunoblotting analysis of tandem-purified WT-R919\* hTRPA1 complexes. FLAG-MBP-WT hTRPA1 and Strep-MBP-R919\* hTRPA1 were transfected separately or together in cells. Lysates were tandem purified for Strep- then FLAG-tagged proteins. MBP-tagged proteins of tandem purification eluates were probed using anti-MBP antibody. In (**a**, **b**, and **e**) Tubulin from whole cell lysates was the loading control. **f** Quantitative analysis of FLAG-MBP-WT hTRPA1 (black) and Strep-MBP-R919\* hTRPA1 (pink) from tandem purifications in **e**. Band intensity was normalized to WT hTRPA1. Data represent mean ± SEM. Means are indicated above the bars. \*\*$p < 0.01$, $n = 3$ independent experiments, two-tailed Student's $t$-test. **a**, **b**, **d**, **e** Full blots are included in Supplementary Fig. 15. **c**, **f** Source data are provided as a Source Data file.

their relative instability and/or revealing transient interactions between R919\* TRPA1 subunits (Supplementary Fig. 7f).

There are three possible stoichiometries for the assembly of WT and R919\* TRPA1 subunits into heterotetramers: 3:1, 2:2, or 1:3 WT:R919\* TRPA1 subunits. To estimate subunit stoichiometry, immunoblot analysis was performed on Strep-FLAG tandem-purified WT-R919\* TRPA1 heteromeric complexes with a common MBP-tag on both subunit types (Fig. 5e). The common MBP-tag readout allows direct comparison of WT and R919\* TRPA1 band intensities within isolated assemblies, yielding an average WT:R919\* TRPA1 subunit ratio of 1:1.3 (Fig. 5f). This subunit ratio is inconsistent with a single WT-R919\* TRPA1 heteromer composition. Together, these data suggest that WT and R919\* TRPA1 subunits co-assemble into stable, tetrameric channel-sized complexes of heterogeneous subunit stoichiometries.

### R919\* TRPA1 subunits contribute to functional channels

Co-assembly of R919\* and WT TRPA1 subunits into channel-sized heteromers raises the possibility that these complexes form functional channels that are a direct cause of the observed hyperactivity. Electrophile agonist complementation assays were performed to investigate whether R919\* TRPA1 subunits reside within functional channels and contribute to activity. AITC activates TRPA1 by covalent modification of three conserved cysteine residues in the cytoplasmic N-terminus (e.g., C621, C641, and C665) which are all present in both WT and R919\* TRPA1 subunits (Figs. 1a and 6a)[11,13]. Mutation of these three cysteines to serine (3CtoS FL) on WT TRPA1 yields a full-length construct with greatly reduced AITC sensitivity (Fig. 6b and

Supplementary Fig. 8a), while preserving activation by the non-electrophile agonist Carvacrol (Supplementary Fig. 8b). To test whether the R919\* mutant subunits contribute to functional channels, the R919\* variant was co-expressed with 3CtoS FL TRPA1 in HEK293T cells and assayed for AITC-evoked activity compared to 3CtoS FL TRPA1 alone. Such experiments revealed that 3CtoS FL TRPA1 AITC sensitivity could be significantly restored by co-expression with the R919\* mutant, suggesting that R919\* subunits can complement the loss of reactive residues (Fig. 6b, compare teal and dark purple bars, Supplementary Fig. 8a, b). This functional rescue was contingent on the R919\* mutant subunits themselves containing the three cysteine residues since co-expression of 3CtoS FL TRPA1 with the 3CtoS R919\* mutant failed to enhance AITC-evoked calcium influx (Fig. 6b, light purple bar, Supplementary Fig. 8a, b). Results from whole-cell voltage clamp experiments in *Xenopus* oocytes were even more striking since a 3CtoA FL TRPA1 variant exhibited almost no AITC-evoked currents at all (Fig. 6c and Supplementary Fig. 8c). Non-electrophile agonist 2-APB did still evoke currents from oocytes expressing 3CtoA FL TRPA1 that were comparable to WT TRPA1 (Supplementary Fig. 8c, g). AITC sensitivity was restored when 3CtoA FL TRPA1 was co-expressed with the R919\* mutant but not when it was co-expressed with the 3CtoA R919\* mutant (Fig. 6d, e and Supplementary Fig. 8d–f). Together, these results support direct co-assembly of WT and R919\* TRPA1 subunits into functional heteromeric channels acting at the plasma membrane with concerted activation among subunits. Moreover, these results demonstrate that electrophile-mediated channel activation can originate from the R919\* mutant subunit(s).

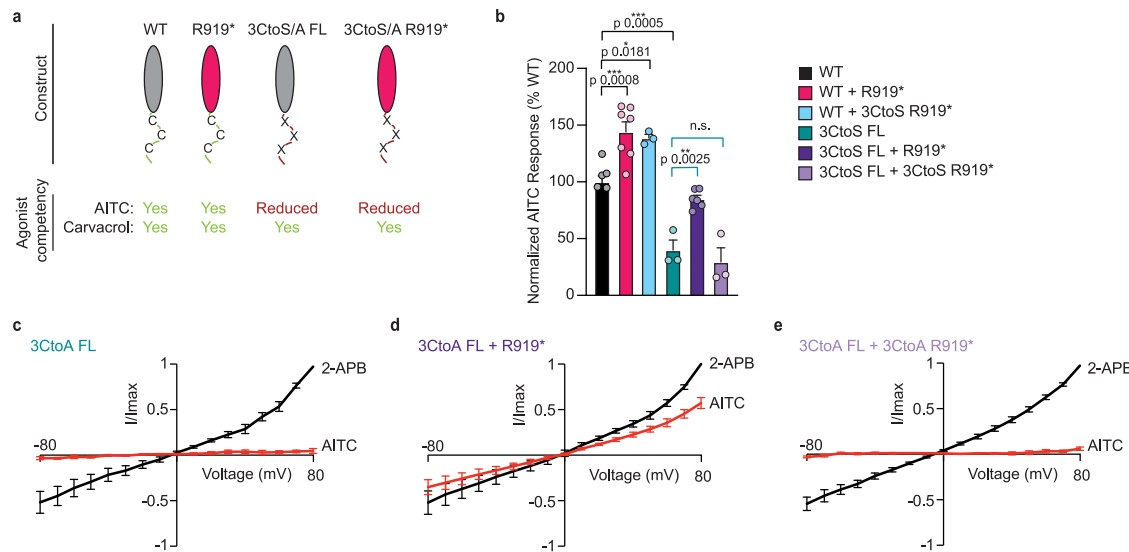

**Fig. 6 | R919* TRPA1 subunits directly contribute to channel activity.**
**a** Schematic of full-length (FL, grey) and R919* (pink) hTRPA1 constructs used for electrophile agonist complementation assays. Agonist competency indicated in green (competent) or red (reduced competency). **b** Quantification of 10 μM AITC-evoked change in Fura-2 ratio relative to maximum response of each expression condition at 100 μM Carvacrol. Expression conditions are WT (black), WT and R919* (pink), WT and 3CtoS R919* (blue), 3CtoS FL (green), 3CtoS FL with R919* (deep purple), and 3CtoS FL with 3CtoS R919* (light purple) hTRPA1. Data further normalized to WT hTRPA1 as 100%. Data represent mean ± SEM. ***p < 0.001, **p < 0.01, *p < 0.05, n.s. not significant. n = 3 (WT with 3CtoS R919*, 3CtoS FL, and 3CtoS FL

with 3CtoS R919* hTRPA1), 5 (WT hTRPA1), 6 (3CtoS FL with R919* hTRPA1), or 7 (WT with R919* hTRPA1) independent experiments, n ≥ 90 cells per transfection condition per experiment, one-way ANOVA with Bonferroni's *post hoc* analysis. Average I-V relationships from *Xenopus* oocytes expressing 3CtoA FL (**c**, n = 7 independent oocytes), 3CtoA FL and R919* (**d**, n = 8 independent oocytes), or 3CtoA FL and 3CtoA R919* (**e**, n = 7 independent oocytes) hTRPA1. Currents were evoked sequentially with 150 μM AITC (red) and 500 μM 2-APB (black) applied to the same oocyte. Currents (I) normalized to 2-APB response at +80 mV (Imax). Extracellular solution contained no calcium. Error bars represent SEM across individual cell measurements. **b–e** Source data are provided as a Source Data file.

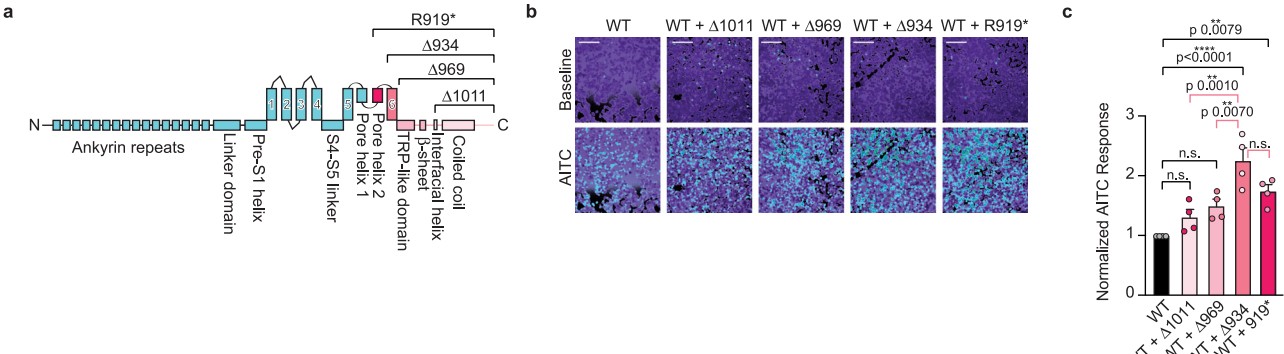

**Fig. 7 | Source of R919* TRPA1-mediated channel hyperactivity. a** Linear diagram depicting major structural domains in a WT hTRPA1 monomer. Missing portions from all C-terminal truncation constructs denoted above. R919* truncation completeness scales with gradations of pink. **b** Ratiometric calcium imaging of HEK293T cells transiently co-transfected with 3×FLAG-WT hTRPA1 and empty vector (mock) or the indicated C-terminal truncation constructs. Cells were stimulated with 10 μM AITC. Scale bars indicate 100 μm. Data representative of four

independent experiments. **c** Quantification of 10 μM AITC-evoked change in Fura-2 ratio relative to maximum response of each expression condition at 100 μM AITC. Data further normalized to WT hTRPA1 response and WT hTRPA1 expression. Colors as indicated in **a**. Data represent mean ± SEM. ****p < 0.0001, **p < 0.01, n.s. not significant. n = 4 independent experiments, n ≥ 60 cells per transfection condition per experiment, one-way ANOVA with Tukey's *post hoc* analysis. Source data are provided as a Source Data file.

## Source of R919* TRPA1-conferred hyperactivity

While AITC sensitivity of 3CtoS or 3CtoA FL TRPA1 could only be restored by R919* TRPA1 subunits with intact cysteines, co-expression of WT TRPA1 with the R919* or 3CtoS R919* mutants conferred similar hyperactivity (Fig. 6b, compare black, pink, and blue bars). These results indicate that the R919* mutant does not require intrinsic agonist sensitivity to modulate WT TRPA1 function and suggest that structural and/or regulatory features lost due to truncation contribute to channel hyperactivity.

TRPA1 structural features that are missing in the R919* mutant can be subdivided into three discrete regions: the second pore helix and S6 transmembrane helix, the latter of which forms

the lower gate; the TRP domain and subsequent β-strand in the membrane-proximal cytoplasmic C-terminus as part of the allosteric nexus important for channel gating; and the remaining cytoplasmic C-terminus including an interfacial helix possibly involved in lipid regulation and the distal coiled coil, which forms the core intracellular inter-subunit interaction and coordinates a requisite polyanion cofactor (Fig. 1a–c)[19,20]. To analyze the contribution of specific functional domains to the conferred channel sensitization, a suite of C-terminal truncation mutants was generated that exclude the coiled coil and interfacial helix (Δ1011–1119), the complete cytoplasmic C-terminus (Δ969–1119), and the S4 transmembrane helix and cytoplasmic C-terminus

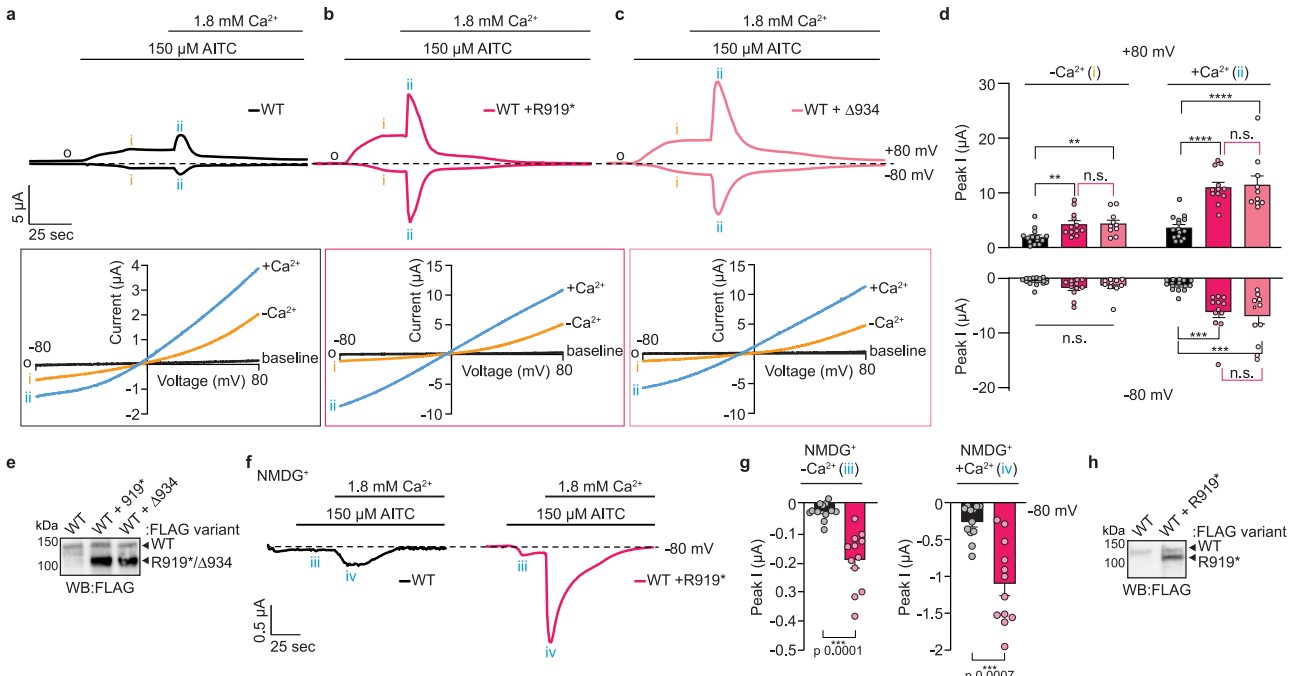

**Fig. 8 | Functional characterization of R919* TRPA1-mediated channel hyper-activity.** Representative time-traces at −80 and +80 mV holding potentials (above) and the corresponding current-voltage relationships from timepoints indicated by o, i and ii (boxed below) from oocytes expressing WT TRPA1 (**a**, black), WT and R919* TRPA1 (**b**, deep pink), or WT and Δ934–1119 hTRPA1 (**c**, pink). Currents in the current-voltage relationships are raw, unadjusted values. Baseline currents (black o) and currents evoked with 150 μM AITC in the absence (orange i) and presence (blue ii) of 1.8 mM extracellular calcium are shown. **d** Quantification of peak current amplitudes from *Xenopus* oocytes used in **a**–**c**. Colors as indicated in **a**–**c**. Data represent mean ± SEM. ****$p < 0.0001$, ***$p < 0.001$, **$p < 0.01$,*$p < 0.05$, n.s. not significant. $n = 10$ (WT with Δ934–1119 hTRPA1), 12 (WT with R919* hTRPA1), or 15 (WT hTRPA1) oocytes per condition, one-way ANOVA with Tukey's *post hoc* analysis. Representative time-traces at −80 mV holding potential (**f**) and peak current amplitude quantification (**g**) from *Xenopus* oocytes expressing WT (black) or WT and R919* (deep pink) hTRPA1 with NDMG[+] as the predominant monovalent extracellular cation. Currents evoked with 150 μM AITC in the absence (iii) and presence (iv) of 1.8 mM extracellular Ca[2+]. Data represent mean ± SEM. ***$p < 0.001$. $n = 12$ oocytes per condition, two-tailed Student's *t*-test. **a**–**c**, **f** Dashed line denotes 0 μA current. Protocol of condition application indicated above. **e**, **h** Western blot of lysates from *Xenopus* oocytes used for representative recordings in **a**–**c** (**e**) and **f** (**h**) expressing 3×FLAG-tagged hTRPA1 variants, probed using HRP-conjugated anti-FLAG antibody. Blots are representative of one oocyte per injection type (WT hTRPA1 ($n = 15$ (**d**) and 12 (**f**)), WT with R919* hTRPA1 ($n = 12$ (**d**, **f**)), and WT with Δ934–1119 hTRPA1 (*n*-10)). Full blots are included in Supplementary Fig. 14. (**a**–**d**, **f** and **g**) Source data are provided as a Source Data file.

(Δ934–1119) (Fig. 7a and Supplementary Fig. 9a, b). Akin to R919* TRPA1, these C-terminal truncations failed to form functional channels when expressed alone in HEK293T cells or *Xenopus* oocytes, suggesting that key regions in the distal cytoplasmic C-terminus are required to support proper channel function (Fig. 1g–i and Supplementary Fig. 9c). This inactivity is seemingly not due to poor expression or failure of the C-terminal truncations to traffic to the plasma membrane (Fig. 1i and Supplementary Fig. 9d). Each C-terminal truncation could additionally immunoprecipitate WT TRPA1 when co-expressed, suggesting that they may also co-assemble with WT subunits into heteromeric complexes (Supplementary Fig. 10a). Ratiometric calcium imaging analysis of WT TRPA1 co-expressed with the partial (Δ1011–1119) and complete (Δ969–1119) cytoplasmic C-terminus truncations revealed a noticeable (though not statistically significant) increase in AITC sensitivity (Fig. 7b, c, light and medium pink bars). However, the additional truncation of the S6 transmembrane helix (Δ934–1119) conferred robust and significant channel hyperactivity when co-expressed with WT TRPA1 (Fig. 7b, c, pink bar). To account for any changes in WT TRPA1 expression for functional experiments, AITC sensitivity was further normalized to WT TRPA1 expression (Supplementary Fig. 10b). These results demonstrate that sequential truncation of the cytoplasmic C-terminus and the S6 transmembrane helix from TRPA1 subunits provide stepwise contributions to R919* TRPA1-conferred channel hyperactivity with loss of the S6 transmembrane helix being largely responsible for heteromeric channel hyperactivation.

## Calcium handling and pore architecture of heteromeric channels

The TRPA1 pore is formed by an S6 transmembrane helix that contains the lower gate and lines the ion permeation pathway as well as an outer pore domain comprised of two extracellular pore helices separated by a loop housing the selectivity filter (Fig. 1a, b). This loop contains an aspartate (D915 in hTRPA1) crucial for calcium permeation and the two pore helices structurally orient the loop along the ion conduction pathway[19,72]. Since the S6 transmembrane helix and the second pore helix are truncated in R919* TRPA1, it is conceivable that WT-R919* heteromeric channels display altered ion selectivity, channel conductance, and/or gating kinetics. Any of these altered properties could account for the observed channel hyperactivity.

To directly assess truncation mutant-mediated changes to channel properties, whole-cell voltage clamp recordings of *Xenopus* oocytes expressing WT TRPA1 with R919* TRPA1 (partially intact outer pore) or Δ934–1119 TRPA1 (fully intact outer pore) were compared in the absence or presence of extracellular calcium. *Xenopus laevis* oocytes endogenously express calcium-activated chloride channels, which could complicate such recordings. However, these channels produce small currents[73,74] and canonical calcium regulation profiles have previously been reported for TRPA1 channels in this heterologous system[24] suggesting endogenous channel contribution is minor and within error. When calcium was excluded from the bath solution, AITC evoked approximately 2-fold larger outward currents (recorded at +80 mV) from oocytes co-expressing WT and R919* or Δ934–1119 TRPA1 compared to those expressing WT TRPA1 alone (Figs. 3a, b and

8a–d). Larger inward currents (recorded at −80 mV) were also typically observed in oocytes co-expressing WT and TRPA1 truncations in calcium-free conditions, though these were not statistically significant (Fig. 8a–d, timepoint i). Calcium causes rapid potentiation of TRPA1 channels[72], which was observed in each channel population (Fig. 8a–c, timepoint ii) and potentiated currents were larger from oocytes co-expressing WT TRPA1 and a truncation construct (Fig. 8d and Supplementary Fig. 10f) despite no significant change to potentiation kinetics (Supplementary Fig. 10g). Specifically, peak outward current amplitudes from oocytes co-expressing WT and R919* or Δ934–1119 TRPA1 were approximately 3-fold larger than from those expressing WT TRPA1 only. Inward peak current amplitudes were also significantly enhanced with 5.3- and 4.8-fold larger currents from oocytes co-expressing WT and R919* or Δ934–1119 TRPA1 than WT TRPA1 only, respectively. These larger currents were not attributed to an increase in WT TRPA1 expression and thus reflect a true increase in channel activity (Fig. 8e).

Calcium potentiation evoked a more linearized current-voltage relationship (e.g., reduced outward rectification) in all TRPA1 expression conditions consistent with significantly increased inward currents (Fig. 8a–c, blue I/V curves, and Supplementary Fig. 10c–e). For WT TRPA1 channels, calcium addition reduced outward rectification from 3.2-fold to 2.7-fold (calculated from $I_{outward}/I_{inward}$). Notably, this reduction in outward rectification with calcium was even more pronounced from oocytes co-expressing WT and R919* (2.4-fold to 1.8-fold upon calcium addition) or Δ934–1119 TRPA1 (3.1-fold to 1.7-fold upon calcium addition), reflective of a larger boost to their inward currents (Fig. 8a–c and Supplementary Fig. 10d, e). Taken together, these data suggest calcium critically contributes to mutant-conferred channel hyperactivity, characterized by significantly larger boosts in calcium-mediated inward currents compared to WT TRPA1 channels.

Following rapid channel potentiation, extracellular calcium triggers slow TRPA1 desensitization[72], which was observed in oocytes expressing WT TRPA1 alone and in those co-expressing WT and R919* or Δ934–1119 TRPA1 (Fig. 8a–c, following timepoint ii). As with potentiation, co-expression of WT TRPA1 with the R919* mutant did not significantly affect desensitization kinetics (Supplementary Fig. 10h). Though not statistically significant, potentiation and desensitization rates were typically faster in oocytes co-expressing WT and R919* TRPA1, suggesting that gating kinetics may be altered in WT-R919* heteromeric TRPA1 channels (Fig. 8a–c and Supplementary Fig. 10g, h). Indeed, the former is consistent with calcium imaging experiments that revealed an enhancement in the time to maximal response at a sub-saturating AITC concentration (Supplementary Fig. 2a). It is likely that more pronounced effects on gating kinetics in the WT-R919* heteromers are masked by the functional co-production of WT TRPA1 homotetramers in these cells. Nonetheless, the observation that channels from oocytes co-expressing WT and R919* TRPA1 or Δ934–1119 TRPA1 exhibit calcium-mediated potentiation and desensitization indicate that these properties are not affected by loss of the S6 transmembrane helix (R919* and Δ934–1119) nor part of the outer pore domain (R919*) in the truncated TRPA1 subunits (Fig. 8a–c). This suggests that elements critical for TRPA1 calcium regulation are either retained in both truncation mutants or that these elements are lost but their presence in the associated WT TRPA1 subunits is sufficient to confer potentiation and desensitization.

The significantly enhanced current amplitudes in oocytes co-expressing WT and R919* TRPA1 raise the possibility that WT-R919* heterochannels exhibit enhanced calcium permeability perhaps due to changes in pore architecture (Fig. 8d). To probe the pore architecture, inward currents were measured with the large monovalent cation N-methyl-D-glucamine (NMDG$^+$) as the predominant extracellular cation. WT TRPA1 typically exhibits low NMDG$^+$ permeability upon acute exposure to agonists[75]. Consistently, WT TRPA1 channels exhibited only small (30 nA) inward currents following activation by AITC while channels from oocytes co-expressing WT and R919* TRPA1 produced significantly larger currents (190 nA) (Fig. 8f, g, timepoint iii), suggesting that the WT-R919* TRPA1 channel pore is larger than that in WT TRPA1 channels. When calcium was added to the extracellular solution, inward currents in both populations increased with significantly larger inward currents from oocytes co-expressing WT and R919* TRPA1 (Fig. 8f, g, timepoint iv). Since NMDG$^+$ permeation was low in oocytes expressing WT or WT and R919* TRPA1, the increase in currents above NMDG$^+$ currents are likely predominantly carried by calcium ions and thus these recordings can provide information on relative calcium permeability. Peak inward currents from oocytes co-expressing WT and R919* were approximately 4-fold higher than those from cells only expressing WT TRPA1 suggesting that WT-R919* TRPA1 channels exhibit enhanced calcium permeability. These larger currents were not attributed to an increase in WT TRPA1 expression in co-injected oocytes and thus reflect a true increase in channel activity (Fig. 8h).

Collectively, these results reveal that channels from oocytes co-expressing WT and R919* TRPA1 exhibit enhanced inward and outward currents especially in the presence of calcium, a more linearized current-voltage relationship with calcium, increased pore size, and enhanced calcium permeability compared to WT TRPA1 alone. These results are consistent with the R919* mutant subunits affecting channel pore architecture and ion permeation. Interestingly, the N855S TRPA1 channelopathy also exhibits enhanced inward currents, however, this property can be detected without calcium (Fig. 1g, h) suggesting the N855S and R919* mutants affect TRPA1 channels through very distinct mechanisms. Endogenously, extracellular environments contain 1.5–2 mM calcium[76] and TRPA1 initiates pain signals by facilitating inward currents to sensory neurons from their resting membrane potential (~−60 to −75 mV)[77]. Thus, the significant enhancement of inward currents introduced by both N855S and R919* TRPA1 mutants should be capable of promoting neuronal excitability.

## Conservation of mutant-conferred hyperactivity and broad applicability

Negative stain electron microscopy of thirteen TRPA1 species orthologues has previously revealed a conserved channel architecture, which raises the possibility that the CRAMPT-associated TRPA1 mutant may similarly confer channel hyperactivity in other orthologues[19]. Analogous CRAMPT-associated mutants were generated for mouse TRPA1 (R922*) and zebrafish TRPA1 isoform a (Q914*), which share 80 and 49% sequence identity to the human TRPA1 orthologue, respectively (Supplementary Fig. 11a). Expression of the mouse R922* or zebrafish Q914* TRPA1 mutants alone in HEK293T cells failed to form functional channels (Supplementary Fig. 11b, c). However, they conferred significantly enhanced AITC sensitivity when co-expressed with their WT TRPA1 counterparts (Fig. 9a, b and Supplementary Fig. 11b, c), suggesting evolutionary conservation of structural features in the S6 transmembrane helix and cytoplasmic C-terminus that contribute to this mechanism of hyperactivity.

To test whether mutant-conferred hyperactivity is conserved across the TRP channel family, truncation mutants analogous to the Δ934–1119 human TRPA1 mutant lacking the S6 transmembrane helix and the cytoplasmic C-terminus were generated for TRPV1 (F656*) and the menthol receptor, TRPM8 (I955*). While TRPA1, TRPV1, and TRPM8 share global structural elements, they possess very structurally distinct outer pore domains and cytoplasmic domains (Supplementary Fig. 12a). Like the analogous Δ934–1119 TRPA1 truncation, neither F656* TRPV1 nor I955* TRPM8 formed functional channels when expressed alone in HEK293T cells despite expressing as well as their WT counterparts (Supplementary Fig. 12b–e). When co-expressed with their WT partners, F656* TRPV1 significantly enhanced capsaicin sensitivity (Fig. 9c and Supplementary Fig. 12b) while the I955* TRPM8 variant had no impact on menthol-evoked activity (Fig. 9d and Supplementary Fig. 12d). Thus, removal of the S6 transmembrane helix

and cytoplasmic C-terminus similarly promotes TRPV1 channel sensitization in TRPV1 heteromers. This is consistent with previously reported enhanced channel activity conferred by a TRPV1 alternative splice variant that removes the part of the cytoplasmic C-terminus responsible for channel regulation by phosphatidylinositols[68,78]. In contrast, removal of these domains failed to confer channel hyperactivity to TRPM8, which is consistent with previous work that demonstrated components of the cytoplasmic C-terminus including the coiled coil are necessary for subunit oligomerization[52,79].

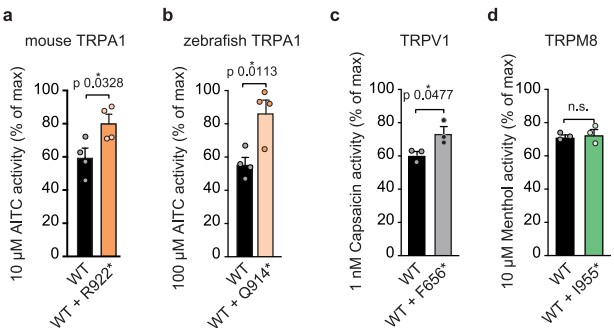

**Fig. 9 | Evolutionary conservation and broad applicability of R919*-conferred channel hyperactivity.** Quantification of sub-saturating agonist-evoked Fura-2 ratio relative to a saturating agonist response for (**a**) mouse TRPA1 variants (10 μM AITC relative to 100 μM AITC), (**b**) zebrafish TRPA1 variants (100 μM AITC relative to 1 mM AITC), (**c**) human TRPV1 variants (1 nM Capsaicin relative to 1 μM capsaicin), and (**d**) rat TRPM8 variants (10 μM Menthol relative to 200 μM Menthol). Colors indicate (**a**) WT (black) or WT with R922* (orange) mouse TRPA1, (**b**) WT (black) or WT with Q914* (light orange) zebrafish TRPA1, (**c**) WT (black) or WT with F656* (grey) human TRPV1, and (**d**) rat TRPM8 variants (10 μM Menthol relative to 200 μM Menthol). Colors indicate (**a**) WT (black) or WT with R922* (orange) mouse TRPA1, (**b**) WT (black) or WT with Q914* (light orange) zebrafish TRPA1, (**c**) WT (black) or WT with F656* (grey) human TRPV1, and (**d**) WT (black) or WT with I955* (green) rat TRPM8. Data represent mean ± SEM. *$p < 0.05$, n.s. not significant. $n = 4$ (**a**, **b**) or 3 (**c**, **d**) independent experiments, $n \geq 90$ cells per transfection condition per experiment, two-tailed Student's $t$-test. Source data are provided as a Source Data file.

Collectively, these results illustrate analogous R919* or Δ934–1119 TRPA1 mutants can be applied to other receptors to confer channel sensitization if their cytoplasmic C-terminal domains are not required for channel assembly.

## Discussion

Gain-of-function channelopathic mutations provide a unique opportunity to understand how changes in channel structure cause aberrant function. Most commonly, gain-of-function channelopathies introduce missense mutations in key structural regions that alter channel gating[18,31,32,80]. Far rarer are truncation channelopathies that lead to gain-of-function, of which only two have been previously characterized[28,30]. In contrast to these variants, CRAMPT-associated R919* *TRPA1* is a drastic nonsense channelopathic mutation that lacks core parts of the ion conduction pathway machinery and the complete cytoplasmic C-terminus. Astonishingly, we found that the R919* mutant conferred gain-of-function when co-expressed with WT TRPA1 subunits. Our data support a model where WT and R919* TRPA1 subunits co-assemble to form hyperactive heteromeric ion channels characterized by enhanced sensitivity to multiple agonist types and by altered pore architecture and increased inward calcium permeation (Fig. 10a). Across neuronal and non-neuronal tissues expressing TRPA1, this increased sensitivity and inward ion conduction could account for the hyperexcitability-hypersensitivity observed in CRAMPT syndrome patients (e.g., cold hyperalgesia, parasthesis, itch, asthma, and GI reflux). Importantly, we found these channels are sensitive to a suite of TRPA1 antagonists, which may illuminate a therapeutic path forward for CRAMPT and other TRPA1 channelopathic disorders.

Our model requires that some WT-R919* TRPA1 heteromeric complexes form functional channels at the plasma membrane (Fig. 10a). Two independent lines of evidence directly support this model. First, R919* TRPA1 restored electrophile agonist sensitivity to full-length TRPA1 subunits lacking the requisite cysteine residues

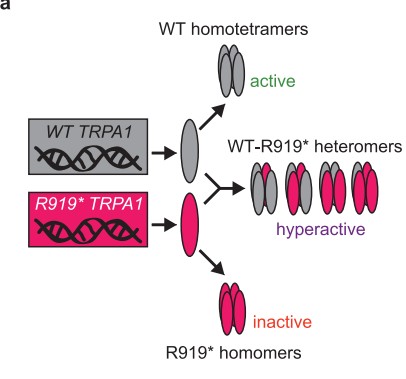

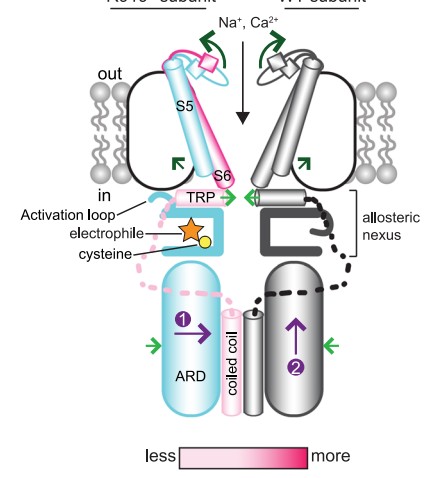

**Fig. 10 | Models of R919*-conferred channel hyperactivity. a** R919* TRPA1 patients are heterozygotes and carry WT and R919* *TRPA1* copies. WT and R919* hTRPA1 protein subunits can assemble into separate homomeric complexes that are active and inactive, respectively. WT and R919* hTRPA1 subunits may also co-assemble into hyperactive hetero-tetrameric channels of four possible subunit stoichiometries and configurations. We propose these heteromeric channels confer gain-of-function. **b** In WT TRPA1 channels, electrophile agonist-evoked gating involves rearrangements in the cytoplasmic domains including conformational flipping of the activation loop, contraction of the coiled coil and adjacent ankyrin repeat domain (ARD), and a sliding rotation of the TRP domain towards the central channel axis (light green arrows). These conformational changes occur with

concerted dilation of the upper and lower gates in the S6 transmembrane helix and selectivity filter, which are coupled through straightening of the S5 transmembrane helix (dark green arrows). Loss of the cytoplasmic C-terminus and S6 transmembrane helix in the R919* mutant contribute to conferred hyperactivity in WT-R919* heteromer channels to different degrees (indicated by gradations of pink). Electrophile modification (orange star) of reactive cysteines (yellow circle) in R919* subunits may communicate channel activation to the pore through associated WT subunits via contraction of the ARD and coiled coil domains (purple arrow 1) that could propagate up to the allosteric nexus (purple arrow 2). Two subunits are shown for clarity.

indicating that R919* subunits are incorporated into these channels (Fig. 6). Second, channels formed in cells co-expressing WT and R919* TRPA1 exhibited a roughly 6-fold increase in current carried by the bulky monovalent cation NMDG[+] compared to WT TRPA1 (Fig. 8f, g). This is consistent with a larger ion channel pore in WT-R919* TRPA1 complexes expected due to the loss of the S6 ion conduction pathway-lining helix from the incorporated R919* subunits. This larger ion channel pore in WT-R919* heteromeric complexes may also account for our observed increase in calcium permeation from cells co-expressing WT and R919* TRPA1.

High-resolution TRPA1 structures solved in distinct functional states allow us to hypothesize how the R919* truncation may enhance channel activity[19–21]. TRPA1 activation is associated with large conformational changes throughout the protein structure including in subdomains lost in the R919* mutant (Fig. 10b, green arrows). These conformational rearrangements may serve as energetic barriers to gating when the channel is in the closed state and loss of key structural elements from R919* subunits may lower the energetic bar for gating in WT-R919* heteromeric complexes by reducing structural constraints that oppose active conformations. Indeed, C-terminal regions have been found to oppose gating in some ion channels[28,81–83]. Consistently, we found that removal of the cytoplasmic C-terminus conferred channel sensitization with a significant boost in activity observed with the additional loss of the S6 transmembrane helix (Fig. 7b, c). This release in structural tension from R919* TRPA1 subunits would be expected to facilitate channel gating in WT-R919* heteromeric complexes originating from cytoplasmic (e.g., AITC and Cinnamaldehyde) or transmembrane (e.g., Carvacrol) ligand binding sites and may explain these channels' enhanced sensitivity to all agonists tested at sub-saturating concentrations. We also show that these truncated C-terminal regions play a similar role in TRPA1 orthologues and in the related TRPV1, suggesting evolutionary conservation of the proposed gating dynamics.

The ability of R919* TRPA1 subunits to initiate electrophile agonist gating of heteromeric channels in our complementation assays suggest channel gating insights (Fig. 6). R919* TRPA1 subunits lack certain key structural features (e.g., the TRP domain and the S6 helix) that communicate this signal to the channel pore (Fig. 10b). However, channel activation is also accompanied by contraction of both the membrane proximal ARD, which forms a cage around the coiled coil, and the coiled coil (Supplementary Fig. 13). Interestingly, coiled coil dynamics have been shown to play a critical role in regulating bacterial voltage-gated sodium channels and TRPM8[81,84]. Though R919* TRPA1 subunits lack the coiled coil helix, contraction of the R919* mutant's ARD may be felt by the coiled coil(s) of associated WT subunits (Fig. 10b, purple arrow 1) and communicated to the pore through their membrane proximal ARD (Fig. 10b, purple arrow 2). Thus, the membrane proximal ARD and coiled coil may represent an intracellular point of intersubunit communication that facilitates concerted channel gating akin to the transmembrane domain. Such a role for ARD dynamics in channel gating has exciting implications for how thermosensation may be mediated by some TRPA1 species orthologues where the ARD has been shown to play a critical role[24,25].

Functional WT-R919* heteromeric complexes appear to retain TRPA1 calcium regulation characterized by rapid potentiation followed by slow desensitization, which raises interesting questions about this important regulatory mechanism (Fig. 8)[72]. To date, four distinct TRPA1 domains have been proposed to contribute to calcium regulation: a N-terminal EF hand, a S2-S3 calcium binding site, a C-terminal acidic cluster, and a C-terminal calmodulin binding domain[21,85–87]. The first two motifs are retained in R919* TRPA1 subunits while the two C-terminal motifs are lost. We observed typical calcium regulation of functional channels in oocytes co-expressing WT and R919* TRPA1 without significant changes to gating kinetics (Fig. 8). This may indicate either that structural features required for calcium-driven potentiation and/or desensitization are retained in the R919* subunits or that associated WT subunits are sufficient to mediate this regulation due to domain-swapped assembly in the transmembrane domain and concerted channel gating[88]. Future work with a homogenous population of WT-R919* TRPA1 complexes will reveal whether and how calcium regulation is affected in these channels.

We found that WT-R919* TRPA1 complexes can form with a mixture of subunit stoichiometries, and future work is needed to determine which of these stoichiometries form in cells to contribute to the CRAMPT syndrome phenotype (Fig. 10a). It is likely that specific WT-R919* TRPA1 heteromer complexes have greatly altered channel properties (e.g., gating kinetics, ion permeation, agonist sensitivity) that are functionally diluted by the co-production of WT homo-tetrameric channels in the same cells in our whole-cell experiments (Fig. 10a). Notably, this heterogenous channel population is likely reflective of the native neuronal environment in CRAMPT syndrome patients and our observed channel sensitization may represent not only an increase in activity of WT-R919* heteromeric complexes, but also a calcium-driven potentiation of co-expressed WT TRPA1 channels. Thus, future work will require production of a homogenous population of WT-R919* TRPA1 complexes with defined subunit stoichiometries and compositions that can be studied in isolation. Such homogeneous preparations will also facilitate structural determination of WT-R919* TRPA1 complexes that could directly reveal how these subunits alter channel architecture to promote sensitization.

The R919* TRPA1 mutant was the first reported genetic link to cramp-fasciculation syndrome (CFS), a rare muscle hyperexcitability syndrome that occurs spontaneously in patients with no family history of the condition. In clinical settings, truncation mutations are often overlooked as they predominantly confer dominant negative or loss-of-function phenotypes[49,89–91]. Thus, our discovery that the R919* TRPA1 mutant confers gain-of-function when co-expressed with WT subunits is noteworthy and distinct. Our findings also expand the possible physiological impact of seemingly nonfunctional truncation mutants, alternative splice variants, or other single nucleotide polymorphisms that are endogenously co-expressed with WT protein and demonstrate that it is critical to consider the genotype of patients where such gene permutations are identified. Finally, our work highlights that spontaneous channelopathic mutations may be more prevalent in the human population than previously appreciated, which provides an impetus for genetic analysis of patients with CFS, CRAMPT, or other stochastic pain syndromes.

## Methods

### Cloning and protein expression

Point mutations to generate the CRAMPT-associated mutant premature stop codon (R919* for human TRPA1, R922* for mouse TRPA1, and Q914* for zebrafish TRPA1a), truncation constructs for TRPV1 (F656*) and TRPM8 (I955*), and electrophile incompetent variants were implemented using QuikChange Lightning site-directed mutagenesis. Larger deletions of C-terminal domains of TRPA1 were accomplished via InFusion EcoDry cloning. For calcium imaging experiments, untagged human TRPA1 in a combined mammalian/oocyte expression vector pMO (modified from pcDNA3 - obtained from David Julius) or 3×FLAG-tagged TRPA1, TRPV1, or TRPM8 in p3×FLAG-eYFP-CMV-7.1 (Addgene #34582) vector were used. WT and inactive human TRPA1 variants were subcloned into CaM/pIRES2-eGFP (Addgene #111499) at the XhoI/EcoRI sites to generate a positive fluorescent readout for transfection in singly transfected calcium imaging studies. For expression in *Xenopus laevis* oocytes, 3×FLAG-hTRPA1 variant genes were subcloned into the combined mammalian/oocyte expression vector pMO prior to generating cRNAs. Further details on plasmid construction can be found in the Supplementary Information. Primers for all constructs used in this study are included in the accompanying Source Data file.

Human embryonic kidney cells (HEK293T, ATCC CRL-3216) were cultured in Dulbecco's modified Eagle's medium (DMEM; Invitrogen) supplemented with 10% calf serum for all experiments except immunofluorescence imaging where 10% FBS was used, and 1x Penicillin-Streptomycin (Invitrogen) at 37 °C and 5% $CO_2$. Cells were grown to ~85–95% confluence before splitting for experiments or propagation.

Plasmids were transfected into HEK293T cells with jetPRIME (Polyplus) according to manufacturer protocols and cells were used for all experiments within 48 h of transfection.

### Ratiometric calcium imaging
After 40–48 h transient transfection, HEK293T cells were loaded with 10 μg/mL Fura-2-acetoxymethylester (Thermo Fisher) in physiological Ringer's buffer (in mM: 120 NaCl, 5 KCl, 2 $CaCl_2$, 25 $NaHCO_3$, 1 $MgCl_2$, 5.5 HEPES, 1 D-glucose, pH 7.4; Boston BioProducts) for ratiometric calcium imaging. Activity of TRPA1 was monitored using a Zeiss Axio Observer 7 inverted microscope with a Hamamatsu Flash sCMOS camera at 20x objective. Dual images (340 and 380 nm excitation, 510 nm emission) were collected and pseudocolour ratiometric images were monitored during the experiment (MetaFluor v7.8.13 software). Following agonist application, cells were imaged for 45–100 s. Details regarding TRPA1 agonist preparation and quantification of ratiometric images can be found in the Supplementary Information.

### Oocyte electrophysiology
pMO vectors carrying 3×FLAG-tagged hTRPA1 constructs were linearized with PmeI, cRNAs were generated by in vitro transcription with the mMessage mMachine T7 transcription kit (Thermo) according to the manufacturer's protocol and were purified with a RNeasy kit (Qiagen). cRNA transcripts were microinjected into surgically extracted *Xenopus laevis* oocytes (Ecocyte) with a Nanoject III (Harvard Apparatus). Oocytes were injected with 0.1 ng (full-length TRPA1 variants) and/or 0.25 ng (truncated variants) of cRNA per cell, and whole-cell currents were measured 24–48 h post-injection using two-electrode voltage clamp (TEVC). Currents were measured using an OC-725D amplifier (Warner Instruments) delivering a ramp protocol from −100 mV to +100 mV applied every second. Microelectrodes were pulled from borosilicate glass capillary tubes and filled with 3 M KCl. Microelectrode resistances of 0.7–1.2 MΩ were used for all experiments. Bath solution contained (in mM) 93.5 NaCl, 2 KCl, 2 $MgCl_2$, 0.1 $BaCl_2$, and 5 HEPES (pH 7.5). For experiments in the presence of calcium, $BaCl_2$ was replaced with 1.8 mM $CaCl_2$. NaCl was replaced with NMDG at equimolar concentrations for experiments in Fig. 8f–h. All recordings were performed at room temperature. Data were subsequently analyzed using pClamp 11 software (Molecular Devices). Internal background currents were subtracted from all recordings unless otherwise indicated. Oocytes were individually collected after recordings, lysed in 50 μL TRPA1 lysis buffer, and subjected to anti-FLAG immunoblot analysis to confirm construct expression as detailed in the Supplementary Information.

### Statistical analysis
All data quantification was performed in Microsoft Excel. Quantified data presentation and statistical analyses were performed in GraphPad Prism. The GraphPad Prism colorblind safe color pallet was applied to all quantified data presented. Data represent mean ± SEM. Each experiment was performed a minimum of three independent times. Comparison between two groups was analyzed by two-tailed Student's *t* test, and comparison between multiple groups was analyzed by one-way ANOVA with Bonferroni or Tukey's *post hoc* analysis, as indicated. Criterion for statistical significance for all tests was $p < 0.05$.

### Reporting summary
Further information on research design is available in the Nature Portfolio Reporting Summary linked to this article.

## Data availability
The data that support the findings of this study in the Supplementary Information are available from the corresponding author upon request. The following PDB files were used in this study 6V9W, 7LP9, 6O6A, and 6V9X. Source data are provided with this paper.

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

## Acknowledgements

We thank Wendy Gilbert, Franziska Bleichert, Elena Gracheva, Joe Howard, Michael Koelle, Christian Schlieker, Tony Koleske, Yong Xiong, Karla Neugebauer, Mark Solomon, David Julius, and Julio Cordero-Morales for constructive suggestions, Fred Sigworth and Youshan Yang for TEVC advice, Grover Paulsen-Sharpe for moral support, and members of the Paulsen lab for helpful discussions and for critical reading of the manuscript. We, additionally, gratefully acknowledge Christian Schlieker for providing the *FLAG-Torsin A* and HA-LBR1600 plasmids. A.P.B. is supported by a Biophysics pre-doctoral training grant (5T32GM008283-33). This research was supported by an International Association for the Study of Pain Early Career Research Grant, a Rita Allen Foundation and American Pain Society Pain Scholar Award, and by NIH grant R35GM142825 to C.E.P., by NIH grant CA244865 to L.K., and by the Pershing Square Sohn Cancer Research Alliance to I.T. and L.K. The content is solely the responsibility of the authors and does not necessarily represent the official views of the National Institutes of Health.

## Author contributions

A.B., S.P.S., and C.E.P. planned the project. A.B., S.P.S., I.T., and C.E.P. designed experiments. A.B., S.P.S., A.L.Z., and C.E.P. carried out calcium imaging and surface biotinylation. A.B. carried out two-electrode voltage clamp experiments. A.B., S.P.S., and C.E.P. carried out pulldowns and proximity biotinylation. A.B. performed Blue Native PAGE. S.P.S. and A.L.Z. carried out fluorescence size exclusion chromatography and crude subunit stoichiometry. I.T. carried out immunostainings. A.B., S.P.S., and C.E.P. wrote the manuscript with input from I.T., A.L.Z., and L.K.

## Competing interests

The authors declare no competing interests.
