## [Peer Review File · Nature Communications]

Molecular Mechanism of Hyperactivation Conferred by a Truncated TRPA1 Disease Mutant Reveals New Gating InsightsReviewers' Comments:

Reviewer #1:

Remarks to the Author:

In this manuscript the authors examine the molecular mechanisms whereby the R919* TRPA1 mutation may cause a clinically-described hypersensitivity-hyperexcitability disorder. They show evidence that the trafficking behavior of the R919* mutant is affected by the presence of WT TRPA1 when the two are coexpressed, as would be expected in patients with heterozygous R919* TRPA1 mutations; they also show evidence that the WT and R919* mutations colocalize at the plasma membrane to form functional, hyperactive heteromeric channels, consistent with a novel mechanism of gain-of-function by a truncation mutant. The findings are of only the third known gain-of-function premature truncation channelopathies, and one that they accurately describe as a "drastic nonsense channelopathic mutation." The findings provide a clear explanation for the (seemingly paradoxical) gain-of-function clinical features of CFS (CRAMPT syndrome). They show in vitro evidence that other TRPs have the potential to also form gain-of-function premature truncation channelopathies as well. The work is thorough, and the manuscript is clearly written. The findings are novel and of considerable clinical and scientific relevance. Specific comments are listed below

Introduction:

- The authors should include the clinical name for the R919* hypersensitivity-hyperexcitability syndrome. It is referred to as CRAMPT syndrome, named for the primary clinical features (cramp-fasciculation syndrome, reflux, asthma/anxiety, migraine, paresthesias/pain, and tachycardia/tremor).
- The reported cases of CRAMPT syndrome were adult-onset (it is not a congenital syndrome like FEPS). This should be corrected in the introduction
- The authors refer vaguely to temperature sensitivity, but in humans TRPA1 is a noxious cold sensor, which should be noted in the introduction. The role of TRPA1 as a noxious cold sensor (responsive specifically to *noxious* cold) fits in well with some of the clinical manifestations of CRAMPT syndrome
- cold hyperalgesia/dysesthesia and cold-sensitive asthma
- In the introduction, the authors could more clearly communicate the importance of this study in investigating why a truncation mutant presents clinically with gain-of-function symptoms. This is well addressed in the discussion, but it might be helpful to set this up more clearly in the introduction
- Regarding the methods: given the clinical responsiveness to carbamazepine, it would be helpful for the authors to evaluate the in vitro response to carbamazepine. In the discussion, the authors might relate their findings to the clinical response to carbamazepine that was reported in the two reported cases of CRAMPT and how this might relate to their findings.
- In the discussion, the authors should consider discussing the implications of their findings relative to the two distinct binding sites for electrophile versus non-electrophile agonists, and how this might relate to the clinical presentation of CRAMPT syndrome.

Reviewer #2:

Remarks to the Author:

This is a provocative ms reporting that a TRPA1 mutant causing cramp-fasciculation syndrome leads to a gain of function exhibiting enhanced activation of the channel by both electrophilic (AITC) and non-electrophilic (carvacrol) agonists. Both pull-down cellular assays and FSEC are suggestive that the mutant and wt subunits interact to form heteromeric channels with altered properties compared to the wt channel. The authors have done a ton of experiments to support the above conclusions, but unfortunately none of the results are completely convincing and I consider them insufficient to advance the core conclusions of the ms when one considers that the mutation leads to truncation of the protein and loss of what are considered essential structural elements, include a pore helix, the pore lining S6 TM helix and the C-terminal domain. For starters, I believe that most of the experiments where calcium imaging where used need to be undertaken with conventional electrophysiological approaches where one can measure currents more directly associated with

opening of the channel. Even if the calcium imaging were supported by electrophysiology, a more likely explanation than heteromeric channel formation would be that expression of the mutant indirectly alters cellular pathways that enhance the activity of TRPA1. However, if the enhanced activation of the channel can be observed using electrophysiology, there would be many additional ways of demonstrating that the presence of the mutant subunit contributes to channel activity by engineering in sensitivity to a chemical, for example introducing a Cys residue that reacts with MTS reagents to inhibit channel function. The FSEC results are intriguing but not sufficiently compelling either. It would be more convincing if the authors introduced spectrally distinct fluorescent proteins into the wt and mutant subunits and demonstrate that both are present before and after purification using SEC. Although one might think that solving a structure of the heteromeric complex would be a ridiculous 'ask', the claims advanced here might be considered sufficiently extraordinary that a structure would be necessary to support them.

Response to referee's comments

We thank the referees for their positive and enthusiastic comments. They have also raised a few important issues that we now address in the revised manuscript, as detailed below in blue:

Response to referee 1:

In this manuscript the authors examine the molecular mechanisms whereby the R919* TRPA1 mutation may cause a clinically-described hypersensitivity-hyperexcitability disorder. They show evidence that the trafficking behavior of the R919* mutant is affected by the presence of WT TRPA1 when the two are coexpressed, as would be expected in patients with heterozygous R919* TRPA1 mutations; they also show evidence that the WT and R919* mutations colocalize at the plasma membrane to form functional, hyperactive heteromeric channels, consistent with a novel mechanism of gain-of-function by a truncation mutant. The findings are of only the third known gain-of-function premature truncation channelopathies, and one that they accurately describe as a “drastic nonsense channelopathic mutation.” The findings provide a clear explanation for the (seemingly paradoxical) gain-of-function clinical features of CFS (CRAMPT syndrome). They show in vitro evidence that other TRPs have the potential to also form gain-of-function premature truncation channelopathies as well. The work is thorough, and the manuscript is clearly written. The findings are novel and of considerable clinical and scientific relevance. Specific comments are listed below:

Introduction:

- The authors should include the clinical name for the R919* hypersensitivity-hyperexcitability syndrome. It is referred to as CRAMPT syndrome, named for the primary clinical features (cramp-fasciculation syndrome, reflux, asthma/anxiety, migraine, paresthesias/pain, and tachycardia/tremor).
- The reported cases of CRAMPT syndrome were adult-onset (it is not a congenital syndrome like FEPS). This should be corrected in the introduction.
- The authors refer vaguely to temperature sensitivity, but in humans TRPA1 is a noxious cold sensor, which should be noted in the introduction. The role of TRPA1 as a noxious cold sensor (responsive specifically to *noxious* cold) fits in well with some of the clinical manifestations of CRAMPT syndrome - cold hyperalgesia/dysesthesia and cold-sensitive asthma.

We thank the referee for these corrections. We have clarified the association of the R919* TRPA1 mutation with CRAMPT syndrome throughout the manuscript, and we now simply refer to FEPS and CRAMPT as TRPA1 disease or channelopathic mutations in the introduction. Upon the referee's suggestion, we have also added 'noxious cold sensor' to the list of roles TRPA1 plays in sensory perception to the introduction.

- In the introduction, the authors could more clearly communicate the importance of this study in investigating why a truncation mutant presents clinically with gain-of-function symptoms. This is well addressed in the discussion, but it might be helpful to set this up more clearly in the introduction.

We thank the referee for this suggestion. We have expanded the final paragraph of the introduction to better contextualize the impact of our findings:

“In clinical settings, it is frequently assumed that pathologies linked to truncations of channel proteins originate from haploinsufficiency or dominant negative effects. However, such broad assumptions have been challenged as more rigorous studies of truncation proteins reveal numerous mechanisms through which they may underlie hyperexcitability-hypersensitivity phenotypes. In this study, we sought to elucidate whether and how the R919* TRPA1 mutant influences channel activity that could explain the observed congenital pain phenotype in patients. Our results expand the functional impact of drastic truncation mutants and provide an impetus for whole exome sequencing of patients with seemingly stochastic pain phenotypes.”

- Regarding the methods: given the clinical responsiveness to carbamazepine, it would be helpful for the authors to evaluate the in vitro response to carbamazepine. In the discussion, the authors might relate their findings to the clinical response to carbamazepine that was reported in the two reported cases of CRAMPT and how this might relate to their findings.

We thank the referee for this suggestion. We assessed whether Carbamazepine (CBZ) affects TRPA1 channels by ratiometric calcium imaging. This new data appears in **Fig. S3**, and we contextualize these findings in the results section:

“Patients harboring the R919* *hTRPA1* mutation experienced symptom relief with the sodium channel blocker Carbamazepine (CBZ) raising the possibility that this molecule directly inhibits TRPA1 channels; however, CBZ failed to affect TRPA1 activity at concentrations below 1 mM (**Fig. S3E-H**). Together, these observations suggest

that co-expression of WT TRPA1 with the R919* mutant yields hyperactive channels that are sensitive to canonical TRPA1 antagonists but not to CBZ, which likely abolishes pain responses downstream of TRPA1.”

- In the discussion, the authors should consider discussing the implications of their findings relative to the two distinct binding sites for electrophile versus non-electrophile agonists, and how this might relate to the clinical presentation of CRAMP T syndrome.

We thank the referee for this suggestion. We have expanded our discussion of our findings that R919* TRPA1 confers enhanced agonist sensitivity independent of agonist type and how this might relate to the clinical presentation of CRAMP T syndrome in the Discussion section.

First: “Our data support a model where WT and R919* TRPA1 subunits co-assemble to form hyperactive heteromeric ion channels characterized by enhanced sensitivity to multiple agonist types and by altered pore architecture and increased inward calcium permeation (**Fig. 7E**). Across neuronal and non-neuronal tissues expressing TRPA1, this increased sensitivity and inward ion conduction could account for the hyperexcitability-hypersensitivity observed in CRAMP T syndrome patients (e.g., cold hyperalgesia, parasthesis, itch, asthma, and GI reflux).”

Second: “This release in structural tension from R919* TRPA1 subunits would be expected to facilitate channel gating in WT-R919* heteromeric complexes originating from cytoplasmic (e.g., AITC and Cinnamaldehyde) or transmembrane (e.g., Carvacrol) ligand binding sites and may explain these channels’ enhanced sensitivity to all agonists tested at sub-saturating concentrations.”

Response to referee 2:

This is a provocative ms reporting that a TRPA1 mutant causing cramp-fasciculation syndrome leads to a gain of function exhibiting enhanced activation of the channel by both electrophilic (AITC) and non-electrophilic (carvacrol) agonists. Both pull-down cellular assays and FSEC are suggestive that the mutant and wt subunits interact to form heteromeric channels with altered properties compared to the wt channel. The authors have done a ton of experiments to support the above conclusions, but unfortunately none of the results are completely convincing and I consider them insufficient to advance the core conclusions of the ms when one considers that the mutation leads to truncation of the protein and loss of what are considered essential structural elements, include a pore helix, the pore lining S6 TM helix and the C-terminal domain. For starters, I believe that most of the experiments where calcium imaging where used need to be undertaken with conventional electrophysiological approaches where one can measure currents more directly associated with opening of the channel.

We thank the referee for this suggestion. In the revised manuscript, we have included many new whole-cell voltage clamp experiments that characterize properties of channels produced in *Xenopus laevis* oocytes co-expressing WT and R919* TRPA1. This heterologous system was used because microinjection of *Xenopus laevis* oocytes allows us to ensure each oocyte has the potential to express both constructs and, for our research questions, is more reliable and experimentally

accessible than the mosaic population achieved by transient co-transfection of HEK293T cells. Such a mosaic population works for ratiometric calcium imaging where you image a field of cells, but it is unmanageable for single cell patch-clamp recordings. Additionally, because our phenomenon centers around co-production and co-assembly of WT and R919* TRPA1 subunits, we built our constructs with an N-terminal 3xFLAG tag, which allowed us to confirm expression of the microinjected constructs from individual oocytes used for data collection. Using this system, we performed two-electrode voltage clamp (TEVC) experiments that supported and expanded on our ratiometric calcium imaging results. We categorize these experiments below:

- 1. Functional assessment of R919*, Δ934-1119, and N855S hTRPA1 in isolation.** We confirmed that the R919* and Δ934-1119 TRPA1 truncation constructs are nonfunctional alone. We also confirmed that the N855S TRPA1 mutant is functional and exhibits a 3.8-fold increase in inward currents consistent with literature precedence. These data are included in **Fig. 1** and we address this in the Results section as:

“HEK293T cells and *Xenopus* oocytes expressing the R919* TRPA1 mutant revealed no activity in the presence of either the electrophile agonist AITC (**Fig. 1D-E** and **1G-H**) and the non-electrophile agonist Carvacrol (**Fig. S1A**), compared to the robust activation observed for WT TRPA1. Additionally, the N855S TRPA1 mutant exhibited enhanced channel activity with a loss of channel rectification and a 3.8-fold increase in inward currents in *Xenopus* whole-cell recordings consistent with its reported gain-of-function properties (**Fig. 1D-E**, **1G-H**, and **S1A**).”

- 2. Assessment of agonist/antagonist sensitivity.** We have found the R919* mutant confers enhanced agonist sensitivity when co-expressed with WT TRPA1, which can be blocked when pre-treated with TRPA1 antagonists. To assess enhance activity by TEVC, we recorded currents from oocytes expressing WT or WT and R919* TRPA1 evoked with 50 and 500 μM AITC. We found both channel populations saturated at 500 μM AITC, but channels from oocytes co-expressing WT and R919* TRPA1 showed significantly larger currents at 50 μM agonist. We additionally showed that already active channels from cells co-expressing WT and R919* hTRPA1 could largely be inhibited by the most selective antagonist, A-967079 by TEVC. These data are included in **Fig. 2** and addressed in the Results and Discussion sections:

Fig. 2. The R919* mutant confers hyperactivity when co-expressed with WT TRPA1. **(C)** Representative I-V relationships from *Xenopus* oocytes expressing WT or WT and R919* hTRPA1. Currents evoked by a sub-saturating (50 μM) and saturating (500 μM) AITC concentrations. Extracellular solution contained 1.8 mM calcium. **(D)** Quantification of 50 μM AITC-evoked peak current amplitudes normalized to 500 μM AITC-evoked currents (max) at +80 mV holding potential. Data represent mean ± SEM. ****p<0.0001. n = 12 oocytes per condition, Student's t-test. **(E)** Representative whole-cell voltage-clamp recordings of *Xenopus* oocytes expressing WT hTRPA1 (black) or WT and R919* hTRPA1 (pink) at +80 mV holding potential. Currents evoked by 150 μM AITC (•) and inhibited by 10 μM A-967079 (§). Dashed line denotes 0 μA current. Protocol of condition application indicated above. Extracellular solution contained no calcium. **(F)** Quantification of AITC-evoked and A-967079-inhibited peak current amplitudes at +80 mV. Data represent mean ± SEM. **p<0.01, n.s. not significant. n = 8-9 oocytes per condition, Student's t-test.

Results: “A similar robust increase in whole-cell currents was observed with a sub-saturating AITC concentration in *Xenopus* oocytes co-expressing WT and R919* TRPA1 compared to those expressing WT protein alone (Fig. 2C-F). These channels were inhibited by the canonical TRPA1 antagonists A-967079, HC-030031, and ruthenium red whether they were applied before or after channel activation, consistent with TRPA1-specific responses in both functional assays (Fig. 2E-F and S3A-D).”

Discussion: Importantly, we found these channels are sensitive to a suite of TRPA1 antagonists, which may illuminate a therapeutic path forward for CRAMPT and other TRPA1 channelopathic disorders.

3. **Assessment of calcium handling and/or regulation.** We observed a significant increase in outward currents from oocytes co-expressing WT and R919* or Δ 934-1119 hTRPA1 in the absence of calcium. We also observed an increase in inward currents, but this was not statistically significant. However, a significant boost to both inward and outward current was observed co-injected populations when we applied extracellular calcium along with a more linearized current-voltage relationship. Interestingly these channels exhibited calcium-mediated potentiation and desensitization with no significant change to kinetics, however, the results were trending towards faster potentiation and desensitization. Recordings with NMDG⁺ as the primary extracellular monovalent cation revealed a larger boost to inward calcium currents from oocytes co-expressing WT and R919* than WT TRPA1 alone. These data are included in Fig. 6 and S10 and addressed in the Results section as follows:

Comparison of peak current amplitudes with or without calcium: “When calcium was excluded from the bath solution, AITC evoked approximately 2-fold larger outward currents (recorded at +80 mV) from oocytes co-expressing WT and R919* or Δ 934-1119 TRPA1 compared to those expressing WT TRPA1 alone (Fig. 2E-F and 6D-G). Larger inward currents (recorded at -80mV) were also typically observed in oocytes co-expressing WT and TRPA1 truncations in calcium-free conditions, though these were not statistically significant (Fig. 6D-G, timepoint i). Calcium causes rapid potentiation of TRPA1 channels⁶⁹, which was observed in each channel population (Fig. 6D-F, timepoint ii) and potentiated currents were larger from oocytes co-expressing WT TRPA1 and a truncation construct (Fig. 6G) despite no significant change to potentiation kinetics (Fig. S10F). Specifically, peak outward current amplitudes from oocytes co-expressing WT and R919* or Δ 934-1119 TRPA1 were approximately 3-fold larger than from those expressing WT TRPA1 only. Inward peak current amplitudes were also significantly enhanced with 5.3- and 4.8-fold larger currents from oocytes co-expressing WT and R919* or Δ 934-1119 TRPA1 than WT TRPA1 only, respectively. These larger currents were not attributed to an increase in WT TRPA1 expression and thus reflect a true increase in channel activity (Fig. 6H).”

Fig. 6. Structural and functional characterization of R919* TRPA1-mediated channel hyperactivity. (D-F) Representative time-traces at -80 and +80 mV holding potentials (above) and the corresponding current-voltage relationships from timepoints indicated by i and ii (boxed below) from oocytes expressing WT TRPA1 (D, black), WT and R919* TRPA1 (E, deep pink), or WT and Δ 934-1119 hTRPA1 (F, pink). Currents in the current-voltage relationships are normalized to +80 mV with calcium. Currents evoked with 150 μM AITC in the absence (i) and presence (ii) of 1.8 mM extracellular calcium. (G) Quantification of peak current amplitudes from *Xenopus* oocytes used in D-F. Data represent mean \pm SEM. **** p <0.0001, *** p <0.001, ** p <0.01, * p <0.05, n.s. not significant. n = 10-15 oocytes per condition, one-way ANOVA with Tukey’s *post hoc* analysis. (H) Western blot of lysates from *Xenopus* oocytes used for recordings in D-F (H) expressing 3xFLAG-tagged hTRPA1 variants, probed using HRP-conjugated anti-FLAG antibody.

Change in rectification in presence of calcium: “Calcium potentiation evoked a more linearized current-voltage relationship (e.g., reduced outward rectification) in all TRPA1 expression conditions consistent with significantly increased inward currents (Fig. 6D-F, blue I/V curves, and Fig. S10C-E). For WT TRPA1 channels, calcium addition reduced outward rectification from 3.2-fold to 2.7-fold (calculated from $I_{\text{outward}}/I_{\text{inward}}$). Notably, this reduction in outward rectification with calcium was even more pronounced from oocytes co-expressing WT and R919* (2.4-fold to 1.8-fold upon calcium addition) or Δ 934-1119 TRPA1 (3.1-fold to 1.7-fold upon calcium addition), reflective of a larger boost to their inward currents (Fig. 6D-F and S10D-E). Taken together, these data suggest calcium critically contributes to mutant-conferred channel hyperactivity, characterized by significantly larger boosts in calcium-mediated inward currents compared to WT TRPA1 channels.”

Fig. S10. Mechanistic dissection of R919*-associated TRPA1 mutant-conferred channel hyperactivity. (C-E) Internal analysis of quantified data from Fig. 6E comparing changes to peak current amplitudes at -80 or +80 mV holding potentials in the absence (filled) and presence (open) of 1.8 mM extracellular calcium for WT hTRPA1 (C), WT and R919* hTRPA1 (D), or WT and D934-1119 hTRPA1 (E). Data represent mean \pm SEM. **** p <0.0001, *** p <0.001, ** p <0.01, * p <0.05. n = 10-15 oocytes per condition, Student's t-test.

Potentiation and desensitization kinetics: “Following rapid channel potentiation, extracellular calcium triggers slow TRPA1 desensitization⁶⁹, which was observed in oocytes expressing WT TRPA1 alone and in those co-expressing WT and R919* or Δ 934-1119 TRPA1 (Fig. 6D-F, following timepoint ii). As with potentiation, co-expression of WT TRPA1 with the R919* mutant did not significantly affect desensitization kinetics (Fig. S10G). Though not statistically significant, potentiation and desensitization rates were typically faster in oocytes co-expressing WT and R919* TRPA1, suggesting that gating kinetics may be altered in WT-R919* heteromeric TRPA1 channels (Fig. 6D-F and S10F-G). Indeed, the former is consistent with calcium imaging experiments that revealed an enhancement in the time to maximal response at a sub-saturating AITC concentration (Fig. S2A). It is likely that more pronounced effects on gating kinetics in the WT-R919* heteromers are masked by the functional co-production of WT TRPA1 homotetramers in these cells. Nonetheless, the observation that channels from oocytes co-expressing WT and R919* TRPA1 or Δ 934-1119 TRPA1 exhibit calcium-mediated potentiation and desensitization indicate that these properties are not affected by loss of the S6 transmembrane helix (R919* and Δ 934-1119) nor part of the outer pore domain (R919*) in the truncated TRPA1 subunits (Fig. 6D-F). This suggests that elements critical for TRPA1 calcium regulation are either retained in both truncation mutants or that these elements are lost but their presence in the associated WT TRPA1 subunits is sufficient to confer potentiation and desensitization.”

Fig. S10. Mechanistic dissection of R919*-associated TRPA1 mutant-conferred channel hyperactivity. (F-G) Calculated time constants of potentiation (F) and desensitization (G) from fitting data from Fig. 6D and E to a single-exponential function. Data represent mean \pm SEM. n.s. not significant. n = 9-15 oocytes per condition, Student's t-test.

NMDG⁺ experiments to assess pore architecture: “The significantly enhanced current amplitudes in oocytes co-expressing WT and R919* TRPA1 raise the possibility that WT-R919* heterochannels exhibit enhanced calcium permeability perhaps due to changes in pore architecture (Fig. 6G). To probe the pore architecture, inward currents were measured with the large monovalent cation *N*-methyl-D-glucamine (NMDG⁺) as the predominant extracellular cation. WT TRPA1 typically exhibits low NMDG⁺ permeability upon acute exposure to agonists⁷⁰. Consistently, WT TRPA1 channels exhibited only small (30 nA) inward currents following activation by AITC while channels from oocytes co-expressing WT and R919* TRPA1 produced significantly larger currents (190 nA) (Fig. 6I-J, timepoint iii), suggesting that the WT-R919* TRPA1 channel pore is larger than that in WT TRPA1 channels. When calcium was added to the extracellular solution, inward currents in both populations increased with significantly larger inward currents from oocytes co-expressing WT and R919* TRPA1 (Fig. 6I-J, timepoint iv). Since NMDG⁺ permeation was low in oocytes expressing WT or WT and R919* TRPA1, the increase in currents above NMDG⁺ currents are likely predominantly carried by calcium ions and thus these recordings can provide information on relative calcium permeability. Peak inward currents from oocytes co-expressing WT and R919* were approximately 4-fold higher

than those from cells only expressing WT TRPA1 suggesting that WT-R919* TRPA1 channels exhibit enhanced calcium permeability. These larger currents were not attributed to an increase in WT TRPA1 expression in co-injected oocytes and thus reflect a true increase in channel activity (Fig. 6K).”

Even if the calcium imaging were supported by electrophysiology, a more likely explanation than heteromeric channel formation would be that expression of the mutant indirectly alters cellular pathways that enhance the activity of TRPA1.

We thank the referee for raising this point. We had previously analyzed cell viability in HEK293T cells expressing WT, R919*, or WT and R919* TRPA1 and saw no effect (Fig. S4D). We also saw no impact of R919* mutant co-expression on WT TRPA1 expression level or plasma membrane localization (Fig. S4A–C and 3A–B). We have now also tested for reactive oxygen species (ROS) production and induction of ER stress, which could be triggered by a truncated protein and affect TRPA1 activity. Here, we also saw no effect. Those new data are included in Fig. S4. Additionally, we previously asked whether co-expression of R919* TRPA1 influenced the related TRPV1 ion channel, which would be expected if the R919* mutant affects cell pathways that sensitize channels. We found the R919* TRPA1 mutant had no effect on TRPV1 activity or agonist sensitivity. These data are included in Fig. S5. Collectively, these results are addressed in the Results section:

“Immunoblot analysis indicated that expression of R919* had no effect on WT TRPA1 levels, even when performed on cells exhibiting hyperactivity (Fig. S4A–C). Similarly, individual or co-expression of WT and R919* TRPA1 had no overt effects on cell viability, reactive oxygen species production, or ER stress induction (Fig. S4D–F). It is possible that R919* TRPA1 causes cell stress not detected by these assays that is capable of generally hypersensitizing ion channels. Thus, R919* TRPA1 was co-expressed with the related capsaicin receptor, TRPV1 and assayed for its effect on human TRPV1 expression, activity, or agonist sensitivity. Akin to WT TRPA1, the R919* mutant had no effect on TRPV1 expression (Fig. S4A–B). Moreover, co-expression with R919* TRPA1 conferred no change in capsaicin evoked TRPV1 calcium influx or agonist sensitivity (Fig. S5). Finally, surface biotinylation assays revealed that co-expression of R919*

had no effect on the amount of WT TRPA1 at the plasma membrane (**Fig. 3A-B**). Together, these data indicate that the R919* mutant does not sensitize channels via changes in WT TRPA1 expression, altered plasma membrane localization, or general induction of cell stress.”

In our heterologous systems, as in the cells of CRAMP syndrome patients carrying the R919* *TRPA1* mutation, WT and R919* TRPA1 protein will be co-produced. Per our model, WT-R919* heteromeric complexes and homotetrameric WT TRPA1 channels may both exist. It is possible and likely that functional WT-R919* heteromeric channels affect the activity of co-produced WT TRPA1 channels perhaps through calcium-dependent regulation. We address this possibility in the Discussion section:

“Notably, this heterogenous channel population is likely reflective of the native neuronal environment in CRAMP syndrome patients and our observed channel sensitization may represent not only an increase in activity of WT-R919* heteromeric complexes, but also a calcium-driven potentiation of co-expressed WT TRPA1 channels. Thus, future work will require production of a homogenous population of WT-R919* TRPA1 complexes with defined subunit stoichiometries and compositions that can be studied in isolation.”

However, if the enhanced activation of the channel can be observed using electrophysiology, there would be many additional ways of demonstrating that the presence of the mutant subunit contributes to channel activity by engineering in sensitivity to a chemical, for example introducing a Cys residue that reacts with MTS reagents to inhibit channel function.

We thank the reviewer for their suggestion. We agree that being able to demonstrate that R919* TRPA1 subunits can uniquely introduce agonist or antagonist sensitivity to WT-R919* TRPA1 complexes would be strong evidence to support their incorporation into functional channels. Three cysteine residues in TRPA1 are targeted by canonical electrophile agonists including AITC and Cinnamaldehyde. We can exploit these electrophile-targeted cysteine residues (C621, C641, and C665) by building insensitive TRPA1 variants wherein these cysteine residues are mutated to serine (3CtoS) or alanine (3CtoA). We previously used ratiometric calcium imaging to show that the R919* mutant, which retains these cysteines could rescue AITC sensitivity to full-length 3CtoS TRPA1 (3CtoS FL) subunits. This rescue was lost when the R919* mutant’s cysteines were also mutated (3CtoS R919*) and these data are presented in **Fig. 5B** and **S8A-B**. In those calcium imaging experiments, the full-length 3CtoS TRPA1 mutant retained some AITC sensitivity. Ideally, we would demonstrate unique introduction of agonist sensitivity by R919* subunits that is otherwise lacking in full-length TRPA1 channels.

We have since performed electrophile agonist complementation experiments in *Xenopus laevis* oocytes with a full-length 3CtoA hTRPA1 (3CtoA FL) construct where we record whole-cell currents and monitor channel activity at the plasma membrane. In this system, the results are even more striking since the full-length 3CtoA TRPA1 channels are completely insensitive to AITC on their own. However, these mutant channels did retain sensitivity to the non-electrophile agonists Carvacrol and 2-APB. Co-expression of the R919* mutant with 3CtoA FL TRPA1 restored AITC-evoked currents that were comparable to 2-APB currents. The ability of the R919* mutant to restore AITC sensitivity was contingent on the R919* subunits having intact cysteines (3CtoA R919*). This R919* mutant-mediated gain in AITC sensitivity was not attributed to a change in full-length 3CtoA TRPA1 expression in oocytes. Moreover, we previously showed R919* subunits do not need their cysteine residues to confer hyperactivity to WT TRPA1 (**Fig. 5B**). These new experiments show that R919* subunits can uniquely introduce agonist sensitivity to functional ion channels at the plasma membrane, and thus, that R919* subunits contribute to channel activity. These new data are presented in **Fig. 5** and **S8**, and they are addressed in the Results section:

Results from whole-cell voltage clamp experiments in *Xenopus* oocytes were even more striking since a 3CtoA FL TRPA1 variant exhibited almost no AITC-evoked currents at all (**Fig. 5C** and **S8C**). Non-electrophile agonists 2-APB and Carvacrol did still evoke currents from oocytes expressing 3CtoA FL TRPA1 that were comparable to WT TRPA1 (**Fig. S8C** and **G**). AITC sensitivity was restored when 3CtoA FL TRPA1 was co-expressed with the R919* mutant but not when it was co-expressed with the 3CtoA R919* mutant (**Fig. 5D-E** and **S8D-F**). Together, these results support direct co-assembly of WT and R919* TRPA1 subunits into functional heteromeric channels acting at the plasma membrane with concerted activation among subunits.”

As discussed in a prior response section, we also probed for functional WT-R919* TRPA1 complexes at the plasma membrane by measuring inward currents when the bulky monovalent cation NMDG⁺ was the predominant extracellular cation. Since R919* subunits lack the second pore helix and the S6 ion conduction pathway-lining transmembrane helix, we would expect that the pore architecture in WT-R919* heteromers will be altered and might better be able to conduct NMDG⁺. Strikingly, we found that co-expression with R919* causes a 6-fold increase in NMDG⁺ current than from WT TRPA1 channels alone indicative of a larger pore. Again, this effect was not associated with a change to WT TRPA1 levels (**Fig. 6K**). These results are presented in **Fig. 6** and are addressed in the Results section:

“To probe the pore architecture, inward currents were measured with the large monovalent cation *N*-methyl-D-glucamine (NMDG⁺) as the predominant extracellular cation. WT TRPA1 typically exhibits low NMDG⁺ permeability upon acute exposure to agonists. Consistently, WT TRPA1 channels exhibited only small (30 nA) inward currents following activation by AITC while channels from oocytes co-expressing WT and R919* TRPA1 produced significantly larger currents (190 nA) (**Fig. 6I-J**, timepoint iii), suggesting that the WT-R919* TRPA1 channel pore is larger than that in WT TRPA1 channels.”

Together, our electrophile agonist complementation assays and recordings with NMDG⁺ provide two pieces of evidence that R919* subunits contribute to functional channels. We contextualize these results in the Discussion section:

“Our model requires that some WT-R919* TRPA1 heteromeric complexes form functional channels at the plasma membrane (**Fig. 7E**). Two independent lines of evidence directly support this model. First, R919* TRPA1 restored electrophile agonist sensitivity to full-length TRPA1 subunits lacking the requisite cysteine residues indicating that R919* subunits are incorporated into these channels (**Fig. 5**). Second, channels formed in cells co-expressing WT and R919* TRPA1 exhibited a roughly 6-fold increase in current carried by the bulky monovalent cation NMDG⁺ compared to WT TRPA1 (**Fig. 6I-J**). This is consistent with a larger ion channel pore in WT-R919* TRPA1 complexes expected due to the loss of the S6 ion conduction pathway-lining helix from the incorporated R919* subunits. This larger ion channel pore in WT-R919* heteromeric complexes may also account for our observed increase in calcium permeation from cells co-expressing WT and R919* TRPA1.”

The FSEC results are intriguing but not sufficiently compelling either. It would be more convincing if the authors introduced spectrally distinct fluorescent proteins into the wt and mutant subunits and demonstrate that both are present before and after purification using SEC.

We thank the reviewer for this suggestion. We have added dual-color FSEC experiments with tandem purified samples from HEK293T cells co-expressing 8xHis-mCerulean-WT hTRPA1 and FLAG-mVenus-R919* hTRPA1. These fluorescent proteins have non-overlapping spectral properties, yet both are present in the same peak from tandem purified samples. This new data is included in **Figure S7** and contextualized in the Results section:

“While fluorescent signal in these [first] experiments was only attributed to one subunit type, multicolor FSEC experiments can be used to characterize heteromeric membrane protein assemblies. Dual color FSEC analysis of Ni²⁺-FLAG tandem-purified complexes from cells co-expressing 8xHis-mCerulean-tagged WT and FLAG-mVenus-tagged R919* TRPA1 revealed an overlapping peak (•) in both the mCerulean and mVenus channels consistent with TRPA1 peaks in Figure 4 (**Fig. S7B-C**). Importantly, mCerulean and mVenus have non-overlapping spectral properties (**Fig. S7D-E**), confirming that the TRPA1 peak detected at both wavelengths from tandem-purified samples must contain both TRPA1 variants.”

Fig. S7. FSEC analysis and Native Blue PAGE of WT and WT-R919* TRPA1 complexes. **(B)** Two-color FSEC chromatogram and **(C)** Western blot analysis of His-FLAG tandem purified complexes from lysates of HEK293T cells co-expressing 8xHis-mCerulean-WT hTRPA1 and FLAG-mVenus-R919* hTRPA1. Tandem purified eluates were split in two and analyzed at the mCerulean (blue) and mVenus (yellow) wavelengths. Peaks corresponding to void tetrameric WT hTRPA1 channels (•) and free FLAG peptide in elution buffer (∞) are indicated. Dashed lines denote the center elution volume of each TRPA1 peak. Samples were also analyzed by Western blot **(C)** using HRP-conjugated anti-FLAG antibody and HRP-conjugated His probe. **(D and E)** FSEC chromatograms of whole cell lysates from HEK293T cells transiently transfected with 8xHis-mCerulean-WT hTRPA1 **(D)** or FLAG-mVenus-R919* hTRPA1 **(E)** analyzed at the mCerulean (blue) and mVenus (yellow) wavelengths. Peaks corresponding to tetrameric WT hTRPA1 channels (•) and free fluorescent protein (∞) are indicated. No spectral contamination was observed between mCerulean and mVenus.

Although one might think that solving a structure of the heteromeric complex would be a ridiculous ‘ask’, the claims advanced here might be considered sufficiently extraordinary that a structure would be necessary to support them.

We thank the reviewer for their suggestion and agree that a WT-R919* TRPA1 structure would be wonderfully informative. This is, however, beyond the scope of the current work since it would be a significant undertaking. Sample heterogeneity, which complicates patch-clamp electrophysiology as outlined above, as well as yield, stability and solution behavior will also complicate solving structures of TRPA1 heteromers with cryo-EM. Future structural work will require production of a homogenous and abundant population of WT-R919* TRPA1 complexes and extensive optimization of cryo-EM sample conditions. We look forward to tackling this for our next story.

Reviewers' Comments:

Reviewer #2:

Remarks to the Author:

I have to say that my skepticism for the core conclusions in this manuscript has begun to crumble and the authors are to be applauded for their courage and persistence. As much as I find the conclusions difficult to swallow, I admit that the substantial amount of additional data provided in the revised version seem to support them. I have some remaining concerns and suggestions that the authors should address.

1) I am dumbfounded by the data shown in Fig.6 where Ca is added to the extracellular solution. Oocytes are well known to contain copious amounts of Ca-activated Cl channels and TRPA1 is Ca permeable, so why aren't the currents measured in external Ca seriously contaminated with Ca-activated Cl currents? The original study cited about regulation by external Ca was done in HEK cells that don't have this issue, but the results the authors show look quite similar even though they shouldn't given the contamination of Ca-activated Cl currents. What is going on here? Have the authors injected BAPTA into the oocytes? Does AITC fortuitously inhibit Ca-activated Cl currents? Did the authors try this experiment with another TRPA1 agonist? This is a critical issue that needs to be dealt with either by providing the necessary background information for why Ca-activated Cl- currents do not contaminate these recordings or to remove this section from the ms.

2) For the many I-V relations shown in the figures, data are not provided for voltage-ramps in the control solution before applying an agonist. Have the authors subtracted these background currents from the I-V relations obtained in the presence of an agonist? It is a good idea to show I-Vs in control solution because it informs on the quality of the recording, but even if these currents have been subtracted, this needs to be stated in the methods.

3) I don't understand what is being plotted in Fig 3B or the explanation on lines 6 and 7 of the legend. Could this be better described? What is the difference between WT + R919* and R919* + WT?

4) In Fig 5 C-E it would be better to separate the data normalized to either 2-APB or Carvacrol and not lump the two agonists used for normalizing together.

5) Fig. S8 doesn't state like the other figures whether the solution contained no Ca. I presume this was the case, but it would be good to state this in each and every figure. Population data are also not provided for Fig. S8A-E and G, which needs to be done.

6) The authors should go through the citations with a fine-tooth comb to make sure there aren't errors. Ref 70 cite in the response to reviewers, for example, has nothing to do with TRPA1 Ca permeability.

Response to referee's comments

We thank the referee for their positive and encouraging comments on our revised manuscript. They have also raised a few important points that we now address in the revised manuscript, as detailed below in blue:

Referee #2:

I have to say that my skepticism for the core conclusions in this manuscript has begun to crumble and the authors are to be applauded for their courage and persistence. As much as I find the conclusions difficult to swallow, I admit that the substantial amount of additional data provided in the revised version seem to support them. I have some remaining concerns and suggestions that the authors should address.

1) I am dumbfounded by the data shown in Fig.6 where Ca is added to the extracellular solution. Oocytes are well known to contain copious amounts of Ca-activated Cl channels and TRPA1 is Ca permeable, so why aren't the currents measured in external Ca seriously contaminated with Ca-activated Cl currents? The original study cited about regulation by external Ca was done in HEK cells that don't have this issue, but the results the authors show look quite similar even though they shouldn't given the contamination of Ca-activated Cl currents. What is going on here? Have the authors injected BAPTA into the oocytes? Does AITC fortuitously inhibit Ca-activated Cl currents? Did the authors try this experiment with another TRPA1 agonist? This is a critical issue that needs to be dealt with either by providing the necessary background information for why Ca-activated Cl- currents do not contaminate these recordings or to remove this section from the ms.

We thank the referee for raising this point. *Xenopus laevis* oocytes endogenously express calcium-activated chloride channels, which could be contributing to our observed currents, however, there are reasons to believe this contribution is minor and does not impact our results and conclusions. Calcium-activated chloride channels have been reported to evoke small ($-0.3 \mu\text{A}$) currents at -80 mV (Boton *et al J Physiol* **1989** 408: 511-534; Leonard and Kelso *Neuron* **1990** 2: 53-60). These currents are much smaller than those evoked from oocytes expressing WT or WT and R919* hTRPA1 in our study, and indeed are within the error of our calcium-containing measurements ($\pm 1 \mu\text{A}$). Thus, the endogenous currents would not be expected to contribute a significant contamination to our measurements. Consistently, a canonical calcium-dependent potentiation and desensitization profile for TRPA1 expressed in *Xenopus laevis* oocytes reported by the Julius group (Cordero-Morales *et al PNAS* **2011** 108: E1184-1191) also showed no signal contamination from endogenous calcium-activated chloride channels.

Despite this precedence and to ameliorate the referee's concern, we measured endogenous calcium-activated chloride channel currents from *Xenopus laevis* oocytes evoked by intrinsic muscarinic acetylcholine receptor activation with Carbachol. As shown below, these channels produced small currents (average of $0.5 \mu\text{A}$ at $+80 \text{ mV}$ and $-0.2 \mu\text{A}$ at -80 mV) with a -20 mV reversal potential, consistent with currents carried by a chloride channel. When scaled to the currents evoked from oocytes expressing WT or WT and R919* hTRPA1 (middle and right panels), these endogenous currents are a small fraction of the total current and reside within the error of our recordings. Consistent with our measured currents being

predominantly carried by the non-selective cation channel TRPA1, the reversal potential in our current-voltage relationships remained near 0 mV in each population both in the absence and presence of calcium.

Referee 2 also raised concerns that AITC may be inhibiting endogenous calcium-activated chloride channels, however, we saw no evidence that AITC affected their activity (teal trace). These currents were partially blocked by the calcium-activated chloride channel blocker DIDS (purple trace).

Collectively, these parameters are consistent with the currents measured in the presence of calcium in this study being predominantly contributed by TRPA1 channels with a small contribution from calcium-activated chloride channels that is within error. We have addressed this concern in the Results Section:

“*Xenopus laevis* oocytes endogenously express calcium-activated chloride channels, which could complicate such recordings. However, these channels produce small currents^{73,74} and canonical calcium regulation profiles have previously been reported for TRPA1 channels in this heterologous system²⁴ suggesting endogenous channel contribution is minor and within error.”

We have not added the above control recordings to Figure S10; however, we will do so if this is deemed necessary.

2) For the many I-V relations shown in the figures, data are not provided for voltage-ramps in the control solution before applying an agonist. Have the authors subtracted these background currents from the I-V relations obtained in the presence of an agonist? It is a good idea to show I-Vs in control solution because it informs on the quality of the recording, but even if these currents have been subtracted, this needs to be stated in the methods.

We thank the reviewer for this suggestion. We internally subtracted baseline currents from most oocytes in this study, and this is now stated in the methods section. We did not subtract internal baseline currents from data in **Fig. 2C**, **Fig. 6D-F**, and **Fig. S8C-E** and **G** where we have added baseline I-V traces for these current-voltage relationships. This is indicated in the figure legend. Moreover, the data in **Fig. S8C-E** are the raw, unadjusted population data for our electrophile agonist complementation assays, which are plotted as average I/Imax traces \pm S.E.M. in **Fig. 5C-E**. Thus, the baseline I/V traces for data in **Fig. 5C-E** are included in **Fig. S8C-E**.

Figure 2. The R919* mutant confers hyperactivity when co-expressed with WT TRPA1. (C) Representative raw I-V relationships from *Xenopus* oocytes expressing WT or WT and R919* hTRPA1. Baseline currents and currents evoked by a sub-saturating (50 μ M) or saturating (500 μ M) AITC concentrations are shown. Extracellular solution contained 1.8 mM calcium.

Figure 6. Structural and functional characterization of R919* TRPA1-mediated channel hyperactivity. (D-F) Representative time-traces at -80 and +80 mV holding potentials (above) and the corresponding current-voltage relationships from timepoints indicated by o, i and ii (boxed below) from oocytes expressing WT TRPA1 (D, black), WT and R919* TRPA1 (E, deep pink), or WT and Δ 934-1119 hTRPA1 (F, pink). Currents in the current-voltage relationships are raw, unadjusted values. Baseline currents (o) and currents evoked with 150 μ M AITC in the absence (i) and presence (ii) of 1.8 mM extracellular calcium are shown.

Figure S8. R919* TRPA1 subunits directly contribute to functional channels. (C-E) Average I-V relationships from *Xenopus* oocytes presented in Fig. 5C-E expressing 3CtoA hTRPA1 (C, n=7), 3CtoA FL and R919* hTRPA1 (D, n=8), or 3CtoA FL and 3CtoA R919* hTRPA1 (E, n=7) showing baseline currents (grey) and sequentially activated with 150 μ M AITC (red) followed by 500 μ M 2-APB (black) in the same oocytes. Data represent mean \pm SEM. (G) Average I-V relationships from *Xenopus* oocytes expressing WT hTRPA1 showing baseline currents (grey) and those evoked by 500 μ M 2-APB (black) in the same oocytes. Data represent mean \pm SEM, n=3. (C-E and G) Extracellular solution contained no calcium.

3) I don't understand what is being plotted in Fig 3B or the explanation on lines 6 and 7 of the legend. Could this be better described? What is the difference between WT + R919* and R919* + WT?

We thank the referee for the question and the opportunity to clarify. **Fig. 3B** shows quantification of surface biotinylation data from **Fig. 3A** for WT (black) or R919* (pink) hTRPA1 expressed alone (solid bars) or co-expressed together (striped bars). We have clarified this data in the figure, figure legend, and the text:

"Finally, surface biotinylation assays revealed that co-expression of R919* had no effect on the amount of WT TRPA1 at the plasma membrane (**Fig. 3A-B**, compare solid and striped black bars)."

"Surface biotinylation assays revealed a small plasma membrane population of the R919* mutant, which increased significantly when co-expressed with WT TRPA1 (**Fig. 3A-B**, compare solid and striped pink bars)."

4) In Fig 5 C-E it would be better to separate the data normalized to either 2-APB or Carvacrol and not lump the two agonists used for normalizing together.

We thank the referee for this suggestion. We have simplified our data analysis to only include oocytes sequentially activated with AITC and 2-APB. The updated data is included in **Fig. 5C-E** and **Fig. S8C-E**:

5) Fig. S8 doesn't state like the other figures whether the solution contained no Ca. I presume this was the case, but it would be good to state this in each and every figure. Population data are also not provided for Fig. S8A-E and G, which needs to be done.

We thank the referee for raising this point. We have indicated in the figure legend for **Fig. S8C-E** and **S8G** that these recordings were, indeed, conducted without extracellular calcium (included above). Additionally, we have added population data for **Fig. S8C-E** and **G**, which can be found above, and **Fig. S8A-B**, as requested:

6) The authors should go through the citations with a fine-tooth comb to make sure there aren't errors. Ref 70 cite in the response to reviewers, for example, has nothing to do with TRPA1 Ca permeability.

We thank the referee for raising this point. We have corrected the indicated reference.

Reviewers' Comments:

Reviewer #2:

Remarks to the Author:

The authors have done a nice job of addressing my concerns and I have no further comments, other than to congratulate the authors for sticking to their views and systematically addressing the many concerns I raised. I will be really interested to see a structure of a heteromeric channel harboring the deletion mutant to understand how this can be a gain of function.